# Budgeted Online Model Selection and Fine-Tuning via Federated Learning

**Pouya M. Ghari**                                        *pmollaeb@uci.edu*
*Department of Electrical Engineering and Computer Science*
*Center for Pervasive Communications and Computing*
*University of California Irvine*

**Yanning Shen**[*]                                       *yannings@uci.edu*
*Department of Electrical Engineering and Computer Science*
*Center for Pervasive Communications and Computing*
*University of California Irvine*

**Reviewed on OpenReview:** *https://openreview.net/forum?id=WeiRR8h87X*

## Abstract

Online model selection involves selecting a model from a set of candidate models 'on the fly' to perform prediction on a stream of data. The choice of candidate models henceforth has a crucial impact on the performance. Although employing a larger set of candidate models naturally leads to more flexibility in model selection, this may be infeasible in cases where prediction tasks are performed on edge devices with limited memory. Faced with this challenge, the present paper proposes an online federated model selection framework where a group of learners (clients) interacts with a server with sufficient memory such that the server stores all candidate models. However, each client only chooses to store a subset of models that can be fit into its memory and performs its own prediction task using one of the stored models. Furthermore, employing the proposed algorithm, clients and the server collaborate to fine-tune models to adapt them to a non-stationary environment. Theoretical analysis proves that the proposed algorithm enjoys sub-linear regret with respect to the best model in hindsight. Experiments on real datasets demonstrate the effectiveness of the proposed algorithm.

## 1 Introduction

The performance of prediction tasks can be heavily influenced by the choice of model. As a result, the problem of model selection arises in various applications and studies, such as reinforcement learning (see, e.g., (Dai et al., 2020; Farahmand & Szepesvari, 2011; Lee et al., 2021)) where a learner selects a model from a set of candidate models to be deployed for prediction. Furthermore, in practical scenarios, data often arrives in a sequential manner, rendering the storage and processing of data in batches impractical. Consequently, there is a need to conduct model selection in an online fashion. In this regard, the learner chooses one model among a *dictionary of models* at each learning round, and after performing a prediction with the chosen model, the learner incurs a loss. The best model in hindsight refers to the model with minimum cumulative loss over all learning rounds while regret is defined as the difference between the loss of the chosen model and the best model in hindsight. Performing model selection online, the goal is to minimize cumulative regret. Depending on the available information about the loss, different online model selection algorithms have been proposed in the literature (see, e.g., (Foster et al., 2017; Muthukumar et al., 2019; Foster et al., 2019; Cella et al., 2021; Pacchiano et al., 2020; Reza Karimi et al., 2021)) which are proven to achieve sub-linear regret. The performance of existing online model selection algorithms depends on the performance of models in the

---

[*]Corresponding author: Yanning Shen. Work in this paper was supported by NSF ECCS 2207457.

dictionary. The choice of dictionary requires information about the performance of models on unseen data, which may not be available a priori. In this case, a richer dictionary of models may improve the performance. However, in practice, the learner may face storage limitations, making storing and operating with a large dictionary infeasible. For example, consider the learner as an edge device performing an online prediction task such as image or document classification using a set of pre-trained models. In this case, the learner may be unable to store all models in the memory. Faced with this challenge, the present work aims at answering the following critical question: *How can learners perform online model selection with a large dictionary of models that requires memory beyond their storage capacity?*

To tackle this problem, the present paper proposes an **o**nline **f**ederated **m**odel **s**election and **f**ine-**t**uning algorithm called OFMS-FT to enable a group of learners to perform online prediction employing a large dictionary of models. To this end, the online model selection is performed in a federated manner in the sense that the group of learners, also known as *clients*, interacts with a server to choose a model among all models stored in the server. Specifically, a server with a significantly larger storage capacity than clients stores a large number of models. At each round, each client chooses to receive and store a subset of models that can be fit into its memory and performs its prediction using one of the stored models. Each client computes the losses of received models and leverages these observed losses to select a model for prediction in future rounds. Moreover, the distribution of data streams observed by clients may differ from the distribution of data that models are pre-trained on. At each round, clients employ the proposed OFMS-FT to adapt models to their data and send the updates to the server. Upon receiving updates from clients, the server updates models.

In addition to the storage capacity of clients, there are some other challenges that should be taken into account. Communication efficiency is a critical issue in federated learning (see e.g.,, (Konečný et al., 2017; Karimireddy et al., 2020; Rothchild et al., 2020; Hamer et al., 2020)). The available bandwidth for client-to-server communication is usually limited which restricts the amount of information that can be sent to the server. To deal with limited communication bandwidth, using the proposed OFMS-FT, at each round the server chooses a subset of clients to fine-tune their received models. Moreover, in some cases, the distribution of data samples may be different across clients (Smith et al., 2017; Zhang et al., 2021a; Li et al., 2021b). Thus, different clients can have different models as the best model in hindsight. Through regret analysis, it is proven that by employing OFMS-FT, each client enjoys sub-linear regret with respect to its best model in hindsight. This shows that OFMS-FT effectively deals with heterogeneous data distributions among clients. In addition, the regret analysis of the present paper proves that the server achieves sub-linear regret with respect to the best model in hindsight. Furthermore, regret analysis shows that increase in communication bandwidth and memory of clients improves the regret bounds. This indicates efficient usage of resources by OFMS-FT. Experiments on regression and image classification datasets showcase the effectiveness of OFMS-FT compared with state-of-the-art alternatives.

## 2 Problem Statement

Consider a set of $N$ clients interacting with a server to perform sequential prediction. There are a set of $K$ models $f_1(\cdot;\cdot),\ldots,f_K(\cdot;\cdot)$ stored at the server. The set of models $f_1(\cdot;\cdot),\ldots,f_K(\cdot;\cdot)$ is called *dictionary* of models. Due to clients' limited memory, clients are not able to store all models $f_1(\cdot;\cdot),\ldots,f_K(\cdot;\cdot)$ and a server with larger storage capacity stores models. Let $[K]$ denote the set $\{1,\ldots,K\}$. At each learning round $t$, client $i$ receives a data sample $\boldsymbol{x}_{i,t}$ and makes a prediction for $\boldsymbol{x}_{i,t}$. Specifically, at learning round $t$, client $i$ picks a model among the dictionary of models $f_1(\cdot;\cdot),\ldots,f_K(\cdot;\cdot)$ and makes prediction $f_{I_{i,t}}(\boldsymbol{x}_{i,t};\boldsymbol{\theta}_{I_{i,t},t})$ where $I_{i,t}$ denote the index of the chosen model by client $i$ at learning round $t$ and $\boldsymbol{\theta}_{k,t}$ represents the parameter of model $k$ at learning round $t$. After making prediction, client $i$, $\forall i \in [N]$ incurs loss $\mathcal{L}(f_{I_{i,t}}(\boldsymbol{x}_{i,t};\boldsymbol{\theta}_{I_{i,t},t}),y_{i,t})$ and observes label $y_{i,t}$ associated with $\boldsymbol{x}_{i,t}$. This continues until the time horizon $T$. The loss function $\mathcal{L}(\cdot,\cdot)$ is the same across all clients and depends on the task performed by clients. For example, if clients perform regression tasks, the loss function $\mathcal{L}(\cdot,\cdot)$ can be the mean square error function while if clients perform classification tasks the loss function $\mathcal{L}(\cdot,\cdot)$ can be chosen to be cross-entropy loss function. Furthermore, the present paper studies the adversarial setting where the label $y_{i,t}$ for any $i$ and $t$ is determined by the environment through a process unknown to clients. This means that the distribution of the data stream $\{(\boldsymbol{x}_{i,t},y_{i,t})\}_{t=1}^{T}$ observed by client $i$ can be non-stationary (i.e., it can change over learning rounds) while client

$i$, $\forall i \in [N]$ does not know the distribution from which $(\boldsymbol{x}_{i,t}, y_{i,t})$, $\forall t$ is sampled. Moreover, data distribution can be different across clients.

The performance of clients can be measured through the notion of regret which is the difference between the client's incurred loss and that of the best model in hindsight. Therefore, the regret of the $i$-th client can be formalized as

$$\mathcal{R}_{i,T} = \sum_{t=1}^{T} \mathbb{E}_t[\mathcal{L}(f_{I_{i,t}}(\boldsymbol{x}_{i,t}; \boldsymbol{\theta}_{I_{i,t},t}), y_{i,t})] - \min_{k \in [K]} \sum_{t=1}^{T} \mathcal{L}(f_k(\boldsymbol{x}_{i,t}; \boldsymbol{\theta}_{k,t}), y_{i,t}), \tag{1}$$

where $\mathbb{E}_t[\cdot]$ denote the conditional expectation given observed losses in prior learning rounds. The objective for each client is to minimize their regret by choosing a model from a dictionary in each learning round. This can be accomplished if clients can identify a subset of models that perform well with the data they have observed. Assessing the losses of multiple models, including the selected one, can help clients better evaluate model performance. This can expedite the identification of models with superior performance, ultimately reducing prediction loss. However, due to memory constraints, clients cannot compute the loss for all models. To address this limitation, clients must select a subset of models that can fit within their memory and calculate the loss for this chosen subset. This information aids in selecting a model for future learning rounds. Furthermore, to enhance model performance, clients and the server collaborate on fine-tuning models. Let $\boldsymbol{\theta}_k^*$ be the optimal parameter for the $k$-th model which is defined as

$$\boldsymbol{\theta}_k^* = \arg\min_{\boldsymbol{\theta}} \sum_{i=1}^{N} \sum_{t=1}^{T} \mathcal{L}(f_k(\boldsymbol{x}_{i,t}; \boldsymbol{\theta}), y_{i,t}). \tag{2}$$

In this context, the regret of the server in fine-tuning model $k$ is defined as

$$\mathcal{S}_{k,T} = \frac{1}{N} \sum_{i=1}^{N} \sum_{t=1}^{T} \mathcal{L}(f_k(\boldsymbol{x}_{i,t}; \boldsymbol{\theta}_{k,t}), y_{i,t}) - \frac{1}{N} \sum_{i=1}^{N} \sum_{t=1}^{T} \mathcal{L}(f_k(\boldsymbol{x}_{i,t}; \boldsymbol{\theta}_k^*), y_{i,t}). \tag{3}$$

The objective of the server is to orchestrate model fine-tuning to minimize its regrets. This paper introduces an algorithmic framework designed to enable clients with limited memory to conduct online model selection and fine-tuning. Regret analysis in Section 4 demonstrates that employing this algorithm leads to sub-linear regret for clients and the server. Additionally, the analysis in Section 4 illustrates that increasing the memory budget for clients results in a tighter upper bound on regret for the proposed algorithm.

## 3 Online Federated Model Selection

The present section introduces a disciplined way to enable clients to pick a model among a dictionary of models beyond the storage capacity of clients while clients enjoy sub-linear regret with respect to the best model in the dictionary. The proposed algorithm enables clients to collaborate with each other to fine-tune models in order to adapt models to clients' data.

### 3.1 Online Federated Model Selection

Let $c_k$ be the cost to store model $k$. For example, $c_k$ can be the amount of memory required to store the $k$-th model. Let $B_i$ denote the budget associated with the $i$-th client, which can be the memory available by the $i$-th client to store models. At each learning round, each client $i$ stores a subset of models that can be fit into the memory of client $i$. The $i$-th client fine-tunes and computes the loss of the stored subset of models. Computing the loss of multiple models at each learning round enables clients to obtain better evaluation on the performance of models which can lead to identifying the best model faster. Moreover, fine-tuning multiple models can result in adapting models to clients' data faster. In Section 4, it is demonstrated that an increase in the number of models that clients can evaluate and fine-tune leads to tighter regret bounds for the proposed algorithm

At learning round $t$, the $i$-th client assigns weight $z_{ik,t}$ to the $k$-th model, which indicates the credibility of the $k$-th model with respect to the $i$-th client. Upon receiving a new data sample, the $i$-th client updates the

weights $\{z_{ik,t}\}_{k=1}^K$ based on the observed losses. The update rule for weights $\{z_{ik,t}\}_{k=1}^K$ will be specified later in equation 8. Using $\{z_{ik,t}\}_{k=1}^K$, at learning round $t$, the $i$-th client selects one of the models according to the probability mass function (PMF) $\boldsymbol{p}_{i,t}$ as

$$p_{ik,t} = \frac{z_{ik,t}}{Z_{i,t}}, \forall k \in [K], \tag{4}$$

where $Z_{i,t} = \sum_{k=1}^K z_{ik,t}$. Let $I_{i,t}$ denote the index of the selected model by the $i$-th client at learning round $t$. Then client $i$ splits all models except for $I_{i,t}$-th model into clusters $\mathbb{D}_{i1,t}, \ldots, \mathbb{D}_{im_iI_{i,t},t}$ such that the cumulative cost of models in each cluster $\mathbb{D}_{ij,t}$ does not exceed $B_i - c_{I_{i,t}}$. Note that $m_{ij,t}$ denote the number of clusters constructed by client $i$ at learning round $t$ if $I_{i,t} = j$. As it will be clarified in Section 4, packing models into minimum number of clusters helps clients to achieve tighter regret bound. Packing models into the minimum number of clusters can be viewed as a bin packing problem (see e.g., (Garey et al., 1972)) which is NP-hard (Christensen et al., 2017). Several approximation methods have been proposed and analyzed in the literature (Johnson, 1973; de la Vega & Lueker, 1981; Dósa, 2007; Hoberg & Rothvoss, 2017). In the present work, models are packed into clusters using the first-fit-decreasing (FFD) bin packing algorithm (see Algorithm 2 in Appendix A). It has been proved that FFD can pack models into at most $\frac{11}{9}m^* + \frac{2}{3}$ clusters where $m^*$ is the minimum number of clusters (i.e., optimal solution for the clustering problem) (Dósa, 2007).

After packing models into clusters, the $i$-th client chooses one of the clusters $\mathbb{D}_{i1,t}, \ldots, \mathbb{D}_{im_iI_{i,t},t,t}$ uniformly at random. Let $J_{i,t}$ be the index of the cluster chosen by client $i$ at learning round $t$. The $i$-th client downloads and stores all models in the cluster $\mathbb{D}_{iJ_{i,t},t}$ along with the $I_{i,t}$-th model. Since the cumulative cost of models in $\mathbb{D}_{iJ_{i,t},t}$ does not exceed $B_i - c_{I_{i,t}}$, the cumulative cost of models in $\mathbb{D}_{iJ_{i,t},t}$ in addition to the $I_{i,t}$-th model does not exceed the budget $B_i$. Let $\mathbb{S}_{i,t}$ represents the subset of models stored by client $i$ at learning round $t$. Upon receiving a new datum at learning round $t$, the $i$-th client performs its prediction task using the $I_{i,t}$-th model. Then, the $i$-th client incurs loss $\mathcal{L}(f_{I_{i,t}}(\boldsymbol{x}_{i,t}; \boldsymbol{\theta}_{I_{i,t},t}), y_{i,t})$. After observing $y_{i,t}$, the $i$-th client computes the loss $\mathcal{L}(f_k(\boldsymbol{x}_{i,t}; \boldsymbol{\theta}_{k,t}), y_{i,t}), \forall k \in \mathbb{S}_{i,t}$. After computing the losses, the $i$-th client obtains the importance sampling loss estimate for the $k$-th model as

$$\ell_{ik,t} = \frac{\mathcal{L}(f_k(\boldsymbol{x}_{i,t}; \boldsymbol{\theta}_{k,t}), y_{i,t})}{q_{ik,t}} \mathcal{I}(k \in \mathbb{S}_{i,t}), \tag{5}$$

where $\mathcal{I}(\cdot)$ denote the indicator function and $q_{ik,t}$ is the probability that the $i$-th client stores the $k$-th model at round $t$. In order to derive $q_{ik,t}$, note that the probability of storing model $k$ at learning round $t$ can be conditioned on $I_{i,t}$ and based on the total probability theorem, it can be obtained that

$$q_{ik,t} = \sum_{j=1}^K \Pr[I_{i,t} = j] \Pr[k \in \mathbb{S}_{i,t} | I_{i,t} = j] = \sum_{j=1}^K p_{ij,t} \Pr[k \in \mathbb{S}_{i,t} | I_{i,t} = j], \tag{6}$$

where $\Pr[\mathcal{A}|\mathcal{B}]$ denote the probability of event $\mathcal{A}$ given that event $\mathcal{B}$ occurred. If client $i$ chooses the $k$-th model at learning round $t$ (i.e., $I_{i,t} = k$), then client $i$ stores model $k$ meaning that $\Pr[k \in \mathbb{S}_{i,t} | I_{i,t} = k] = 1$. Let $m_{ij,t}$ denote the number of clusters when client $i$ chooses $I_{i,t} = j$. Note that if $I_{i,t} = j$ client $i$ splits all models except for model $j$ into $m_{ij,t}$ clusters. If client $i$ chooses $I_{i,t} = j$ such that $j \neq k$, then client $i$ stores model $k$ at learning round $t$ if the chosen cluster contains model $k$. Since only one cluster among $m_{ij,t}$ clusters contains model $k$ and client $i$ chooses one cluster uniformly at random, it can be concluded that $\Pr[k \in \mathbb{S}_{i,t} | I_{i,t} = j, j \neq k] = \frac{1}{m_{ij,t}}$. Therefore, according to equation 6, the probability $q_{ik,t}$ can be obtained as

$$q_{ik,t} = p_{ik,t} + \sum_{\forall j : j \neq k} \frac{p_{ij,t}}{m_{ij,t}}. \tag{7}$$

At learning round $t$, client $i$ updates the weights of models using the multiplicative update rule as

$$z_{ik,t+1} = z_{ik,t} \exp\left(-\eta_i \ell_{ik,t}\right), \tag{8}$$

where $\eta_i$ is the learning rate associated with the $i$-th client. According to equation 5 and equation 8, if client $i$ does not observe the loss of model $k$ at learning round $t$, client $i$ keeps the weight $z_{ik,t+1}$ the same as $z_{ik,t}$. Conversely, when the loss is observed, a higher loss corresponds to a greater reduction in the weight $z_{ik,t}$.

### 3.2 Online Model Fine-Tuning

This subsection presents a principled way to enable clients to collaborate with the server to fine-tune models. If client $i$ participates in fine-tuning at learning round $t$, client $i$ locally updates all its stored models in $\mathbb{S}_{i,t}$ and sends updated models' parameters to the server. Sending updated parameters of a model to the server occupies a portion of communication bandwidth. Therefore, the available communication bandwidth may not be enough such that all clients can send their updates every learning round. In this case, at learning round $t$, the server has to choose a subset of clients $\mathbb{G}_t$ such that the available bandwidth is sufficient for all clients in $\mathbb{G}_t$ to send their updates to the server.

Let $b_k$ be the bandwidth required to send the updated parameters of the $k$-th model. If $i \in \mathbb{G}_t$, then the $i$-th client needs the bandwidth $e_i = \sum_{k \in \mathbb{S}_{i,t}} b_k$ to send updated parameters of its stored models. Let the available communication bandwidth be denoted by $E$. Given $e_i$, $\forall i \in [N]$, the server splits clients into $\alpha$ groups $\mathbb{N}_1, \ldots, \mathbb{N}_\alpha$ such that $\sum_{i \in \mathbb{N}_j} e_i \leq E, \forall j \in [\alpha]$. This means that all clients in each group $\mathbb{N}_j$, $\forall j \in [\alpha]$ can send their updated model parameters to the server given the available bandwidth. At each learning round $t$, the server draws one of the client groups $\mathbb{N}_1, \ldots, \mathbb{N}_\alpha$ uniformly at random. Clients in the chosen group send their updated models' parameters to the server. In other words, $\mathbb{G}_t := \mathbb{N}_{\iota_t}$ where $\iota_t$ represents the index of the chosen client group at learning round $t$. Since the probability of choosing a client group is $\frac{1}{\alpha}$, the probability that a client is chosen by the server to send its updated models parameters is $\frac{1}{\alpha}$ as well. Therefore, to maximize the probability that a client is chosen by the server to be in $\mathbb{G}_t$, the server can split clients into a minimum number of groups. To do this, bin-packing algorithms such as FFD (Dósa, 2007) can be employed.

Define the importance sampling loss gradient estimate $\nabla \hat{\ell}_{ik,t}$ associated with client $i$ and model $k$ as

$$\nabla \hat{\ell}_{ik,t} = \frac{\alpha}{q_{ik,t}} \nabla \mathcal{L}(f_k(\boldsymbol{x}_{i,t}; \boldsymbol{\theta}_{k,t}), y_{i,t}) \mathcal{I}(i \in \mathbb{G}_t, k \in \mathbb{S}_{i,t}). \tag{9}$$

Recall that $\mathcal{I}(\cdot)$ denote the indicator function. The $i$-th client updates the models' parameters as

$$\boldsymbol{\theta}_{ik,t+1} = \boldsymbol{\theta}_{k,t} - \eta_f \nabla \hat{\ell}_{ik,t}, \forall k \in \mathbb{S}_{i,t}, \tag{10}$$

where $\eta_f$ is the fine-tuning learning rate. According to equation 9 and equation 10, if the $i$-th client is not chosen for fine-tuning (i.e., $i \notin \mathbb{G}_t$), then $\boldsymbol{\theta}_{ik,t+1} = \boldsymbol{\theta}_{k,t}$. Thus, if $i \in \mathbb{G}_t$, the $i$-th client sends locally updated models parameters $\{\boldsymbol{\theta}_{ik,t+1}\}_{\forall k \in \mathbb{S}_{i,t}}$ to the server. Let $\mathbb{V}_{k,t}$ denote the index set of clients such that $i \in \mathbb{V}_{k,t}$ if the server receives the update of the $k$-th model $\boldsymbol{\theta}_{ik,t+1}$ from the $i$-th client. Upon aggregating information from clients, the server updates model parameters $\{\boldsymbol{\theta}_{k,t}\}_{k=1}^K$ as:

$$\boldsymbol{\theta}_{k,t+1} = \boldsymbol{\theta}_{k,t} - \frac{1}{N} \sum_{i \in \mathbb{V}_{k,t}} (\boldsymbol{\theta}_{k,t} - \boldsymbol{\theta}_{ik,t+1}) = \boldsymbol{\theta}_{k,t} - \frac{\eta_f}{N} \sum_{i=1}^N \nabla \hat{\ell}_{ik,t}. \tag{11}$$

The proposed Online Federated Model Selection and Fine-Tuning (OFMS-FT) algorithm is summarized in Algorithm 1. Steps 5 to 10 in Algorithm 1 outline how client $i$ selects a subset of models during each learning round $t$. Subsequently, each client $i$, $\forall i \in [N]$, transmits its required bandwidth $e_i$ (acquired in step 10) to the server. Upon receiving $e_i$, $\forall i \in [N]$, the server, following steps 12 to 14, determines a subset of clients $\mathbb{G}_t$ eligible to send their updates. Each client $i$ then utilizes its selected model for prediction (step 16) and computes the loss of its stored model subset, updating the weights $z_{ik,t}{}_{k=1}^K$ (step 17). If client $i$ is included in the server's selected subset $\mathbb{G}_t$, it transmits its local updates to the server, as depicted in step 19. Finally, in step 22, the server updates the model parameters for use in the subsequent learning round.

## 4 Regret Analysis

The present section analyzes the performance of OFMS-FT in terms of cumulative regret. To analyze the performance of OFMS-FT, it is supposed that the following assumptions hold:
**(a1)** For any $(\boldsymbol{x}, y)$ and $\boldsymbol{\theta}$, the loss is bounded as $0 \leq \mathcal{L}(f_k(\boldsymbol{x}; \boldsymbol{\theta}), y) \leq 1$.
**(a2)** The budget satisfies $B_i \geq c_k + c_j$, $\forall k, j \in [K]$, $\forall i \in [N]$.

---
**Algorithm 1** OFMS-FT: Online Federated Model Selection and Fine-Tuning

---
1: **Input:** Models $f_k(\cdot; \boldsymbol{\theta}_{k,0})$, costs $c_k$, $b_k$ and budgets $B_i$, $E$, $\forall i, \forall k$.
2: **Initialize:** $z_{ik,1} = 1$, $\forall k, \forall i$.
3: **for** $t = 1, \ldots, T$ **do**
4:      **for all** $i \in [N]$, the $i$th client **do**
5:          Chooses a model using PMF $\boldsymbol{p}_{i,t}$ in equation 4. $I_{i,t}$ denote the index of the chosen model.
6:          Splits all models except for $I_{i,t}$-th model into clusters $\mathbb{D}_{i1,t}, \ldots, \mathbb{D}_{im_{i,t},t}$ such that

$$\sum_{k \in \mathbb{D}_{ij,t}} c_k \leq B_i - c_{I_{i,t}}, \forall j : 1 \leq j \leq m_{i,t}.$$

7:          Chooses one cluster among $\{\mathbb{D}_{ij,t}\}_{j=1}^{m_{i,t}}$ uniformly at random where $J_{i,t}$ is the chosen cluster index.
8:          Constructs the set of model indices $\mathbb{S}_{i,t} = \{I_{i,t}\} \cup \{k | \forall k \in \mathbb{D}_{iJ_{i,t},t}\}$.
9:          Downloads all models whose indices are in $\mathbb{S}_{i,t}$ from the server.
10:         Sends the bandwidth cost $e_i = \sum_{k \in \mathbb{S}_{i,t}} b_k$ to the server.
11:      **end for**
12:      The server splits clients into $\alpha$ groups $\mathbb{N}_1, \ldots, \mathbb{N}_\alpha$ such that $\sum_{i \in \mathbb{N}_j} e_i \leq E, \forall j \in [\alpha]$.
13:      The server draws one of the groups $\{\mathbb{N}_j\}_{j=1}^\alpha$ uniformly at random.
14:      The server finds $\mathbb{G}_t := \mathbb{N}_{\iota_t}$ with $\iota_t$ as the chosen client group index.
15:      **for all** $i \in [N]$, the $i$th client **do**
16:          Makes prediction $f_{I_{i,t}}(\boldsymbol{x}_{i,t}; \boldsymbol{\theta}_{I_{i,t},t})$ and computes $\mathcal{L}(f_k(\boldsymbol{x}_{i,t}; \boldsymbol{\theta}_{k,t}), y_{i,t})$, $\forall k \in \mathbb{S}_{i,t}$.
17:          Updates $z_{ik,t}$, $\forall k \in \mathbb{S}_{i,t}$ according to equation 8.
18:          **if** $i \in \mathbb{G}_t$ **then**
19:              Sends $\boldsymbol{\theta}_{ik,t+1}$, $\forall k \in \mathbb{S}_{i,t}$ obtained by equation 10 to the server.
20:          **end if**
21:      **end for**
22:      The server updates models parameters as in equation 11.
23: **end for**

---

**(a3)** The loss function $\mathcal{L}(f_k(\boldsymbol{x}; \boldsymbol{\theta}), y)$ is convex with respect to $\boldsymbol{\theta}$, $\forall k \in [K]$.
**(a4)** The gradient of the loss function is bounded as $\|\nabla \mathcal{L}(f_k(\boldsymbol{x}; \boldsymbol{\theta}), y)\| \leq G$, $\forall \boldsymbol{\theta}$, $\forall k \in [K]$. Also, $\boldsymbol{\theta}$ belongs to a bounded set such that $\|\boldsymbol{\theta}\|^2 \leq R$.

Let $m_{ij}^*$ be the minimum number of clusters if client $i$ splits all models except for model $j$ such that the cumulative cost of each cluster does not exceed $B_i - c_j$. Define $\mu_i = \max_j m_{ij}^*$, which can be interpreted as the upper bound for the minimum number of clusters that can be constructed by client $i$ at each learning round. It can be concluded that $\mu_i < K$ and increase in budget $B_i$ leads to decrease in $\mu_i$. The following theorem obtains the upper bound for the regret of clients and the server in terms of $\mu_i$.

**Theorem 1.** *Assume that clients utilize the first fit decreasing algorithm in order to split models into clusters $\mathbb{D}_{i1,t}, \ldots, \mathbb{D}_{im_{i,t},t}$. Under (a1) and (a2), the expected cumulative regret of the $i$-th client using OFMS-FT is bounded by*

$$\mathcal{R}_{i,T} \leq \frac{\ln K}{\eta_i} + \eta_i \mu_i T, \tag{12}$$

*which holds for all $i \in [N]$. Under (a1)–(a4), the cumulative regret of the server in fine-tuning model $k$ using OFMS-FT is bounded by*

$$\mathcal{S}_{k,T} \leq \frac{R}{2\eta_f} + \frac{1}{N} \sum_{i=1}^{N} \mu_i \alpha \eta_f G^2 T. \tag{13}$$

*Proof.* see Appendix B.        $\square$

If the $i$-th client sets $\eta_i = \mathcal{O}\left(\sqrt{\frac{\ln K}{\mu_i T}}\right)$, then the $i$-th client achieves sub-linear regret of

$$\mathcal{R}_{i,T} \leq \mathcal{O}\left(\sqrt{(\ln K)\mu_i T}\right). \tag{14}$$

If the server sets the fine-tuning learning rate as $\eta_f = \frac{1}{\sqrt{\frac{\alpha T}{N}\sum_{i=1}^{N}\mu_i}}$, then the server achieves regret of

$$\mathcal{S}_{k,T} \leq \mathcal{O}\left(\sqrt{\frac{\alpha T}{N}\sum_{i=1}^{N}\mu_i}\right). \tag{15}$$

The regret bounds in equation 14 and equation 15 show that a decrease in $\mu_i$ leads to tighter regret bound. If the $i$-th client has a larger budget $B_i$, the upper bound for the minimum number of model clusters $\mu_i$ decreases since client $i$ can split models into clusters with larger budgets. As an example, consider the special case that all models have the same cost $c_k = c$, $\forall k \in [K]$ while $B_i = \delta_i c$ where $\delta_i \geq 2$ is an integer. In this case, according to step 6 in Algorithm 1, the upper bound for the minimum number of clusters is $\mu_i = \frac{(K-1)c}{(\delta_i-1)c} = \frac{K-1}{\delta_i-1}$. Therefore, as the budget of a client increases, the client can achieve tighter regret bound. In addition, larger communication bandwidth enables the server to partition clients into smaller number of groups $\alpha$. Thus, according to equation 15, larger communication bandwidth leads to tighter regret bound for the server.

**Challenges of Obtaining Regret in equation 15.** Limited memory of clients brings challenges for obtaining the sub-linear regret in equation 15 since clients cannot calculate the gradient loss of all models every learning round. To overcome this challenge, the present paper proposes update rules equation 9 and equation 10 along with the novel model subset selection presented in steps 5, 6 and 7 of Algorithm 1. In what follows the effectiveness of the proposed update rules and model subset selection is explained. Employing vanilla online gradient descent update rule $\boldsymbol{\theta}_{ik,t+1} = \boldsymbol{\theta}_{k,t} - \nabla\mathcal{L}(f_k(\boldsymbol{x}_{i,t}; \boldsymbol{\theta}_{k,t}), y_{i,t})$ locally by clients, can result in regret of $\mathcal{O}(\sqrt{T})$ if client $i$ knows $\nabla\mathcal{L}(f_k(\boldsymbol{x}_{i,t}; \boldsymbol{\theta}_{k,t}), y_{i,t})$, $\forall k \in [K]$ at every learning round (see e.g., (Hazan, 2022)). This is not possible since client $i$ has limited memory and is not able to calculate the gradient loss of all models every learning round. To overcome this challenge, the present paper proposes the local update rules in equation 9 and equation 10 which use the gradient loss estimate $\nabla\hat{\ell}_{ik,t}$ instead of true loss gradient. Using equation 37 of Appendix B, it can be concluded that $\nabla\hat{\ell}_{ik,t}$ is an unbiased estimator of $\nabla\mathcal{L}(f_k(\boldsymbol{x}_{i,t}; \boldsymbol{\theta}_{k,t}), y_{i,t})$ meaning that $\mathbb{E}_t[\nabla\hat{\ell}_{ik,t}] = \nabla\mathcal{L}(f_k(\boldsymbol{x}_{i,t}; \boldsymbol{\theta}_{k,t}), y_{i,t})$. According to equation 9, obtaining $\nabla\hat{\ell}_{ik,t}$ does not require storing model $k$ and calculating $\nabla\mathcal{L}(f_k(\boldsymbol{x}_{i,t}; \boldsymbol{\theta}_{k,t}), y_{i,t})$ every learning round. Hence, $\nabla\hat{\ell}_{ik,t}$ can be obtained every learning round given the limited memory of client $i$. However, according to equation 36, equation 38 and equation 39 of Appendix B, employing update rule of equation 10 the regret of the server grows with $\mathbb{E}_t[\|\nabla\hat{\ell}_{ik,t}\|^2]$ which is upper bounded as $\mathbb{E}_t[\|\nabla\hat{\ell}_{ik,t}\|^2] \leq \frac{\alpha G^2}{q_{ik,t}}$ (see equation 37b in Appendix B). Therefore, the regret of the server grows with $\frac{1}{q_{ik,t}}$ where $q_{ik,t}$ is the probability that client $i$ stores model $k$ and fine-tunes it at learning round $t$. Using model subset selection method presented in steps 5, 6 and 7 of Algorithm 1, the probability $q_{ik,t}$ is obtained as in equation 7. In equation 29 of Appendix B, it is proven that if clients employ FFD algorithm to cluster models in step 6, then it is guaranteed that $q_{ik,t} \geq \frac{1}{2\mu_i}$. This lead to guaranteeing the regret upper bound in equation 15.

## 5 Related Works and Discussions

This Section discusses differences, innovations, and improvements provided by the proposed OFMS-FT compared to related works in the literature.

**Online Model Selection.** Online model selection algorithms by Foster et al. (2017); Muthukumar et al. (2019); Foster et al. (2019); Cella et al. (2021); Pacchiano et al. (2020); Reza Karimi et al. (2021) have studied either *full-information* or *bandit* settings. Full-information refers to cases where the loss of all models can be observed at every round while in bandit setting only the loss of the chosen model can be observed. Regret bounds obtained by full-information based online model selection algorithms cannot be guaranteed if the

learner (i.e., a client) cannot store all models. Moreover, it is useful to mention that the present paper studies the adversarial setting where the losses observed by clients at each round are specified by the environment and may not follow any time-invariant distribution. It is well-known that in the adversarial bandit setting the learner achieves regret upper bound of $\mathcal{O}(\sqrt{KT})$ (see e.g., (Pacchiano et al., 2020)). The proposed OFMS-FT utilizes the available memory of clients to evaluate a subset of models every round (see step 16 in Algorithm 1) which helps client $i$ to achieve regret of $\mathcal{O}(\sqrt{\mu_i T})$ as presented in equation 14. If client $i$ is able to store more than one model, then $\mu_i < K$ (see below equation 3 and the discussion below Theorem 1) which shows that OFMS-FT utilizes the available memory of clients to improve their regret bound compared to bandit setting. Moreover, aforementioned online model selection works have not studied online fine-tuning of models. A model selection algorithm proposed by Pacchiano et al. (2022) assumes that each model (called base learner) comes with a candidate regret bound and utilizes this information for model selection. By contrast, the present paper assumes that there is no available prior information about the performance of models. Moreover, Muthukumar & Krishnamurthy (2022) has studied the problem of model selection in linear contextual bandits where the reward (can be interpreted as negative loss in our work) is the linear function of the context (can be interpreted as $\boldsymbol{x}_{i,t}$ in our work). However, the present paper does not make this assumption in both theoretical analysis and experiments. Furthermore, online model selection when models are kernels has been studied in the literature (see e.g., (Yang et al., 2012; Zhang et al., 2021b; Li & Liao, 2022; Ghari & Shen, 2023a)) where the specific characteristics of kernel functions are exploited to perform model selection to alleviate computational complexity of kernel learning.

**Online Learning with Partial Observations.** Another line of research related to the focus of the present paper is online learning with expert advice where a learner interacts with a set of experts such that at each learning round the learner makes decision based on advice received from the experts (Cesa-Bianchi & Lugosi, 2006). The learner may observe the loss associated with a subset of experts after decision making, which can be modeled using a graph called feedback graph (Mannor & Shamir, 2011; Amin et al., 2015; Cohen et al., 2016; Alon et al., 2015; Cortes et al., 2020; Ghari & Shen, 2023b). In online federated model selection, each model can be viewed as an expert. Employing the proposed OFMS-FT, in addition to performing online model selection, clients and the server collaborate to fine-tune the models (experts). However, the aforementioned online learning algorithms do not study the case where the learner can influence experts. Performing online model selection and fine-tuning jointly in a federated fashion brings challenges for guarantying sub-linear regret that cannot be overcame using the existing online learning algorithms. Specifically, due to limited client-to-server communication bandwidth and limited memory of clients, all clients are not able to fine-tune all models every learning round. The proposed OFMS-FT introduces a novel model subset selection in steps 5, 6 and 7 of Algorithm 1 and a novel update rule in equation 9 and equation 10 to fine-tune models locally by clients in such a way that given limited memory of clients and limited communication bandwidth, the server achieves sub-linear regret of equation 15.

**Online Federated Learning.** The problem of online federated model selection and fine-tuning is related to online federated learning (Chen et al., 2020; Mitra et al., 2021; Damaskinos et al., 2022). Chen et al. (2020) has studied learning a global model when clients receive new data samples while they participate in federated learning. Online decision-making by clients has not been studied by Chen et al. (2020) and hence the regret bound for clients cannot be guaranteed. An online federated learning algorithm has been proposed by Damaskinos et al. (2022) to cope with the staleness in federated learning. However, it lacks theoretical analysis when clients need to perform online decision-making. An online mirror descent-based federated learning algorithm called Fed-OMD has been proposed in Mitra et al. (2021). Fed-OMD obtains sub-linear regret when clients perform their online learning task while collaborating with the server to learn a single model. However, Fed-OMD cannot guarantee sub-linear regret when it comes to performing online model selection if clients are unable to store all models in the dictionary. Furthermore, Hong & Chae (2022); Gogineni et al. (2022); Ghari & Shen (2022) have studied the problem of online federated learning where each client learns a kernel-based model employing specific characteristics of kernel functions.

**Personalized Model Selection and Fine-Tuning.** In addition to online model selection, online learning and online federated learning discussed in Section 5, personalized federated learning can be related to the focus of this paper. Employing the proposed OFMS-FT, model selection and fine-tuning is personalized for clients. According to step 5 in Algorithm 1, each client chooses a model locally using the personalized

PMF $\boldsymbol{p}_{i,t}$ in equation 4 to make a prediction at round $t$. This helps each client $i$, $\forall i \in [N]$ to achieve the sub-linear regret in equation 14. Furthermore, the choice of models to be fine-tuned locally by each client is personalized according to step 19 in Algorithm 1. Particularly, $q_{ik,t}$ in equation 7 is the probability that the client $i$ fine-tunes the model $k$ at round $t$. The probability $q_{ik,t}$ is determined by client $i$ and it can be inferred that the probability to participate in fine-tuning a model is determined by client $i$ based on its preferences given the limited memory budget. It is useful to add that personalized federated learning is well-studied topic related to the focus of the present paper. In personalized federated learning framework, aggregating information from clients the server assists clients to learn their own personalized model. Several personalized federated learning approaches have been proposed in the literature for example inspired by model-agnostic meta-learning (Finn et al., 2017; Fallah et al., 2020; Acar et al., 2021), adding a regularization term to the objective function (Hanzely et al., 2020; Dinh et al., 2020; Li et al., 2021a; Liu et al., 2022), among others (see e.g., (Deng et al., 2020; Collins et al., 2021; Marfoq et al., 2021; Shamsian et al., 2021)). However, none of the aforementioned works have studied online decision making, online federated model selection and fine-tuning when clients have limited memory and employing them, sub-linear regrets cannot be guaranteed for clients and the server.

**Client Selection.** Client selection in federated learning has been extensively explored in the literature (Chen et al., 2018; Huang et al., 2021; Balakrishnan et al., 2022; Németh et al., 2022; Fu et al., 2023). However, none of the aforementioned works have specifically studied client selection for online federated learning, where clients utilize the trained model from federated learning for online predictions. In the proposed OFMS-FT, the server selects clients for their participation in model fine-tuning uniformly at random. This choice aims to avoid differentiating among clients and fine-tune models in the favor of any clients. Nevertheless, an intriguing direction for future research is to investigate how alternative client selection strategies, beyond uniform selection, could enhance client regret in the context of online federated learning.

## 6 Experiments

We tested the performance of the proposed OFMS-FT for online model selection through a set of experiments. The performance of OFMS-FT is compared with the following baselines: MAB (Auer et al., 2003), Non-Fed-OMS, RMS-FT, B-Fed-OMFT, FedOMD (Mitra et al., 2021) and PerFedAvg (Fallah et al., 2020). MAB refers to the case where the server chooses a model using Exp3 algorithm (Auer et al., 2003) and transmits the chosen model to all clients. Then, each client sends the loss of the received model to the server. Non-Fed-OMS refers to **non-fed**erated **o**nline **m**odel **s**election where each client stores a fixed subset of models that can be fit into its memory. At each learning round, each client chooses one model from the stored subset of models using Exp3 algorithm. RMS-FT denote a baseline where at each learning round each client chooses a subset of models uniformly at random to fine-tune them. The prediction task is then carried out by selecting one of the chosen models uniformly at random. Furthermore, B-Fed-OMFT stands for Budgeted Federated Online Model Fine-Tuning. In this approach, the server maintains a set of models that can be fit into the memory of all clients. Clients collaborate with the server to fine-tune all models in each learning round. In the B-Fed-OMFT framework, each client employs the Exp3 algorithm to choose one model to perform the prediction task. Using Fed-OMD (Mitra et al., 2021), given a pre-trained model, clients and the server fine-tune the model. PerFedAvg refers to the case where given a pre-trained model, clients and the server fine-tune the model using Personalized FedAvg (Fallah et al., 2020). The performance of the proposed OFMS-FT and baselines are tested on online image classification and online regression tasks. Image classification tasks are performed over CIFAR-10 (Krizhevsky, 2009) and MNIST (Lecun et al., 1998) datasets. Online regression tasks are performed on Air (Zhang et al., 2017) and WEC (Neshat et al., 2018) datasets. For both image classification datasets, the server stores 20 pre-trained convolutional neural networks (CNNs). Based on the number of parameters required to store models, for CIFAR-10 the normalized costs of storing CNNs are either 0.89 or 1, and for MNIST the normalized costs are either 0.66 or 1. The experiments were conducted with a single meta-replication, utilizing a consistent random seed for both the proposed OFMS-FT and all baseline methods. Moreover, for both regression datasets, the server stores 20 pre-trained fully-connected feedforward neural networks. Since all neural networks have the same size, the normalized costs of all of them are 1. More details about models and datasets can be found in Appendix C.

Table 1: Average and standard deviation of clients' accuracy over CIFAR-10 and MNIST datasets. MSE ($\times 10^{-3}$) and its standard deviation ($\times 10^{-3}$) across clients over Air and WEC datasets.

| Algorithms | CIFAR-10 | MNIST | Air | WEC |
|---|---|---|---|---|
| MAB | $61.14\% \pm 10.12\%$ | $84.37\% \pm 5.18\%$ | $8.97 \pm 6.82$ | $36.96 \pm 8.52$ |
| Non-Fed-OMS | $65.76\% \pm 12.24\%$ | $85.06\% \pm 7.30\%$ | $8.82 \pm 6.70$ | $64.84 \pm 40.30$ |
| RMS-FT | $57.86\% \pm 9.75\%$ | $88.33\% \pm 4.25\%$ | $7.87 \pm 5.22$ | $18.89 \pm 4.02$ |
| B-Fed-OMFT | $67.83\% \pm 12.28\%$ | $89.46\% \pm 4.71\%$ | $8.09 \pm 5.55$ | $31.94 \pm 7.97$ |
| Fed-OMD | $64.34\% \pm 9.89\%$ | $88.69\% \pm 5.16\%$ | $11.37 \pm 7.16$ | $30.89 \pm 10.06$ |
| PerFedAvg | $55.65\% \pm 11.94\%$ | $89.71\% \pm 4.93\%$ | $11.29 \pm 7.08$ | $30.09 \pm 10.17$ |
| OFMS-FT | $\mathbf{76.77\% \pm 4.46\%}$ | $\mathbf{92.05\% \pm 2.69\%}$ | $\mathbf{7.46 \pm 5.10}$ | $\mathbf{7.09 \pm 1.67}$ |

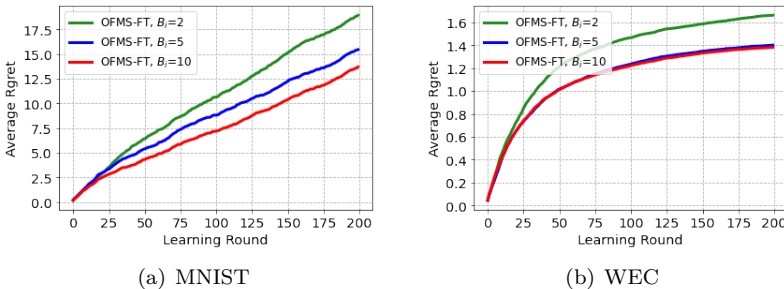

(a) MNIST           (b) WEC

Figure 1: Average regret of clients using OFMS-FT with the change in budget $B_i$.

There are 50 clients performing image classification task, and 100 clients performing online regression task. Note that clients are performing the learning task in an online fashion such that at each learning round each client observes one data sample and predicts its label in real time. The learning rates $\eta_i$ for all methods are set to be $10/\sqrt{T}$ where $T = 200$. Furthermore, the fine-tuning learning rate is set to $\eta_f = 10^{-3}/\sqrt{T}$. Since the required bandwidth to send a model is proportional to the model size, the normalized bandwidth $b_k$ associated with the $k$-th model are considered to be the same as the normalized cost $c_k$.

Table 1 demonstrates the average and standard deviation of clients' accuracy over CIFAR-10 and MNIST when the test set is distributed in non-i.i.d manner among clients. For CIFAR-10, each client receives 155 testing data samples from one class and 5 samples from each of the other nine classes. In the case of MNIST, each client receives at least 133 samples from one class and at least 5 samples from the other classes. The 200 testing data samples are randomly shuffled and are sequentially presented to each client over $T = 200$ learning rounds. The accuracy of the client $i$ is defined as $\text{Accuracy}_i = \frac{1}{T} \sum_{t=1}^{T} \mathcal{I}(\hat{y}_{i,t} = y_{i,t})$ where $\hat{y}_{i,t}$ denote the class label predicted by the algorithm. The memory budget is $B_i = 5, \forall i \in [N]$. Each client is able to store up to 50 images. In order to fine-tune models, clients employ the last 50 observed images. Moreover, Fed-OMD and PerFedAvg fine-tune one of the 20 pre-trained models such that both fine-tune the same pre-trained model. As can be observed from Table 1, the proposed OFMS-FT achieves higher accuracy than Non-Fed-OMS and B-Fed-OMFT. This corroborates that having access to larger number of models helps learners to achieve better learning performance, especially in cases where the learners are faced with heterogeneous data, and performance of models are unknown in priori. Moreover, from Table 1 it can be seen that OFMS-FT achieves higher accuracy than MAB which admits the effectiveness of observing losses of multiple models at each learning round. Higher accuracy of OFMS-FT compared with Fed-OMD and PerFedAvg indicates the benefit of fine-tuning multiple models rather than one. The superior performance of OFMS-FT in comparison to RMS-FT highlights the efficacy of employing model selection through the PMF defined in equation equation 4, as opposed to choosing models uniformly at random. In addition, as can be seen from Table 1, the standard deviation of clients' accuracy associated with OFMS-FT is considerably lower than other baselines. This shows that using OFMS-FT, accuracy across clients shows less variations

compared with other baselines. Therefore, the results confirm that OFMS-FT can cope with heterogeneous data of clients in more flexible and henceforth more effective fashion.

Furthermore, Table 1 presents the mean square error (MSE) of online regression and its standard deviation across clients for Air and WEC datasets. Specifically, MSE of client $i$ is defined as $\text{MSE}_i = \frac{1}{T} \sum_{t=1}^{T} (\hat{y}_{i,t} - y_{i,t})^2$. In both the Air and WEC datasets, individual data samples are associated with one of four geographical areas. The distribution of data samples across clients is non-i.i.d, with 50 clients observing data samples from one specific site, while the remaining 50 clients observe data samples from another geographical site. At each round, only half of clients are able to send their updated models to the server. All other settings are the same as online image classification setting. Results for online regression tasks are consistent with the conclusions obtained from the results of online image classification task. Moreover, Figure 1 illustrates the average regret of clients using the proposed OFMS-FT through learning rounds for different values of memory budget $B_i$. Figure 1 depicts the regret of OFMS-FT through learning on MNIST and WEC datasets. As can be seen, the increase in $B_i$ leads to obtaining lower regret by clients.

Therefore, the results in Figure 1 are in agreement with the regret analysis in Section 4. Table 2 illustrates the sensitivity of the MSE and its standard deviation, as achieved by the proposed OFMS-FT, to the budget $B_i$ ($\forall i \in [N]$) over the WEC dataset. In this configuration, the budget varies across clients, with a subset having $B_i = 3$ and the remainder $B_i = 5$. The improvement in MSE becomes evident as the number of clients with a budget of $B = 5$ increases. This observation indicates that an increase in budget enhances the performance of OFMS-FT, aligning with the theoretical findings presented in Section 4.

Table 2: MSE ($\times 10^{-3}$) and its standard deviation ($\times 10^{-3}$) of OFMS-FT across clients over WEC dataset under varying budgets among clients.

| $B_i = 3$ | $B_i = 5$ | MSE |
|---|---|---|
| 60% | 40% | $7.96 \pm 1.49$ |
| 50% | 50% | $7.85 \pm 1.66$ |
| 40% | 60% | $7.32 \pm 1.54$ |

## 7 Conclusion

Performing online model selection with a large number of models can improve the performance of online model selection especially when there is not prior information about models. The present paper developed a federated learning approach (OFMS-FT) for online model selection when clients cannot store all models due to limitations in their memory. To adapt models to clients' data, employing OFMS-FT clients can collaborate to fine-tune models. It was proved that both clients and the server achieve sub-linear regret with respect to the best model in hindsight. Experiments on regression and image classification datasets were carried out to showcase that the proposed OFMS-FT achieved better performance in comparison with non-federated online model selection approach and other state-of-the-art federated learning algorithms which employ a single model rather than a dictionary of models.

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

## A First Fit Decreasing Algorithm

In this paper, first fit decreasing algorithm is employed to split models into clusters. To begin with, order models by decreasing cost. Let $s(1), \ldots, s(K-1)$ denote the indices of all models except for the $I_{i,t}$-th model ordered in ascending manner according to models' costs such that if $i \leq j$, then $c_{s(i)} \leq c_{s(j)}$. At the $k$-th step of clustering, client $i$ checks whether the $s(k)$-th model can be fit into any currently existing clusters according to budget $B_i - c_{I_{i,t}}$. The $s(k)$-th model is put into the first cluster that it can be fit into. Otherwise, if it cannot be fit into any opened cluster, then it is assigned to a new cluster indexed by $m_{i,t} + 1$. This continues until all models except for the $I_{i,t}$-th model are corresponded to a cluster. Algorithm 2 summarizes the clustering procedure performed by the $i$-th client.

---

**Algorithm 2** Cluster Generation by Client $i$ at Learning Round $t$

---

1: **Input:** Chosen model index $I_{i,t}$, costs $c_k$, $\forall k \in [K]$ and budget $B_i$.
2: **Initialize:** $m_{i,t} = 1$.
3: Order all models except for the $I_{i,t}$-th model by decreasing cost to obtain $s(1), \ldots, s(K-1)$.
4: **for** $k = 1, \ldots, K-1$ **do**
5:     Set $j = 1$ and $d = 0$
6:     **while** $d = 0$ and $j \leq m_{i,t}$ **do**
7:         **if** $\sum_{m \in \mathbb{D}_{ij,t}} c_m + c_{s(k)} \leq B_i - c_{I_{i,t}}$ **then**
8:             Set $d = 1$ and $j = j + 1$
9:         **end if**
10:     **end while**
11:     **if** $j = m_{i,t} + 1$ and $d = 0$ **then**
12:         Assign the $s(k)$-th model to a new cluster $\mathbb{D}_{i(m_{i,t}+1),t}$.
13:         Update $m_{i,t} = m_{i,t} + 1$.
14:     **end if**
15: **end for**
16: **Output:** $\{\mathbb{D}_{i1,t}, \ldots, \mathbb{D}_{im_{i,t},t}\}$

---

## B Proof of Theorem 1

In order to prove Theorem 1, the following Lemma is used as the step-stone.

**Lemma 2.** *Under (a1) and (a2), the regret of the $i$-th client with respect to any model $k$ is bounded from above as*

$$\sum_{t=1}^{T} \mathbb{E}_t[\mathcal{L}(f_{I_{i,t}}(\boldsymbol{x}_{i,t}; \boldsymbol{\theta}_{I_{i,t},t}), y_{i,t})] - \sum_{t=1}^{T} \mathcal{L}(f_k(\boldsymbol{x}_{i,t}; \boldsymbol{\theta}_{k,t}), y_{i,t}) \leq \frac{\ln K}{\eta_i} + \eta_i \mu_i T. \tag{16}$$

*Proof.* Recall that $Z_{i,t} = \sum_{k=1}^{K} z_{ik,t}$. Therefore, we can write

$$\frac{Z_{i,t+1}}{Z_{i,t}} = \sum_{k=1}^{K} \frac{z_{ik,t+1}}{Z_{i,t}} = \sum_{k=1}^{K} \frac{z_{ik,t}}{Z_{i,t}} \exp\left(-\eta_i \ell_{ik,t}\right). \tag{17}$$

According to equation 4, $\frac{z_{ik,t}}{Z_{i,t}} = p_{ik,t}$ and as a result equation 17 can be rewritten as

$$\frac{Z_{i,t+1}}{Z_{i,t}} = \sum_{k=1}^{K} p_{ik,t} \exp\left(-\eta_i \ell_{ik,t}\right). \tag{18}$$

Combining the inequality $e^{-x} \leq 1 - x + \frac{1}{2}x^2, \forall x \geq 0$ with equation 17 we can conclude that

$$\frac{Z_{i,t+1}}{Z_{i,t}} \leq \sum_{k=1}^{K} p_{ik,t} \left(1 - \eta_i \ell_{ik,t} + \frac{1}{2}(\eta_i \ell_{ik,t})^2\right). \tag{19}$$

Employing the inequality $1 + x \leq e^x$ and taking logarithm from both sides of equation 19, we arrive at

$$\ln \frac{Z_{i,t+1}}{Z_{i,t}} \leq \sum_{k=1}^{K} p_{ik,t} \left( -\eta_i \ell_{ik,t} + \frac{1}{2}(\eta_i \ell_{ik,t})^2 \right). \tag{20}$$

Summing equation 20 over learning rounds leads to

$$\ln \frac{Z_{i,T+1}}{Z_{i,1}} \leq \sum_{t=1}^{T} \sum_{k=1}^{K} p_{ik,t} \left( -\eta_i \ell_{ik,t} + \frac{1}{2}(\eta_i \ell_{ik,t})^2 \right). \tag{21}$$

In addition, $\ln \frac{Z_{i,T+1}}{Z_{i,1}}$ can be bounded from below as

$$\ln \frac{Z_{i,T+1}}{Z_{i,1}} \geq \ln \frac{z_{ik,T+1}}{Z_{i,1}} = -\eta_i \sum_{t=1}^{T} \ell_{ik,t} - \ln K, \tag{22}$$

which holds for any $k \in [K]$. Combining equation 21 with equation 22, we get

$$\sum_{t=1}^{T} \sum_{k=1}^{K} p_{ik,t} \ell_{ik,t} - \sum_{t=1}^{T} \ell_{ik,t} \leq \frac{\ln K}{\eta_i} + \frac{\eta_i}{2} \sum_{t=1}^{T} \sum_{k=1}^{K} p_{ik,t} \ell_{ik,t}^2. \tag{23}$$

Considering equation 5 and equation 7, given the observed losses in prior rounds the expected value of $\ell_{ik,t}$ can be obtained as

$$\mathbb{E}_t[\ell_{ik,t}] = \frac{\mathcal{L}(f_k(\boldsymbol{x}_{i,t}; \boldsymbol{\theta}_{k,t}), y_{i,t})}{q_{ik,t}} p_{ik,t} + \sum_{\forall j: j \neq k} \frac{\mathcal{L}(f_k(\boldsymbol{x}_{i,t}; \boldsymbol{\theta}_{k,t}), y_{i,t})}{q_{ik,t}} \frac{p_{ij,t}}{m_{ij,t}}$$

$$= \frac{\mathcal{L}(f_k(\boldsymbol{x}_{i,t}; \boldsymbol{\theta}_{k,t}), y_{i,t})}{q_{ik,t}} \left( p_{ik,t} + \sum_{\forall j: j \neq k} \frac{p_{ij,t}}{m_{ij,t}} \right) = \mathcal{L}(f_k(\boldsymbol{x}_{i,t}; \boldsymbol{\theta}_{k,t}), y_{i,t}). \tag{24}$$

Moreover, based on the assumption that $0 \leq \mathcal{L}(f_k(\boldsymbol{x}_{i,t}; \boldsymbol{\theta}_{k,t}), y_{i,t}) \leq 1$, given the observed losses in prior rounds, the expected value of $\ell_{ik,t}^2$ can be bounded from above as

$$\mathbb{E}_t[\ell_{ik,t}^2] = \frac{\mathcal{L}^2(f_k(\boldsymbol{x}_{i,t}; \boldsymbol{\theta}_{k,t}), y_{i,t})}{q_{ik,t}^2} \left( p_{ik,t} + \sum_{\forall j: j \neq k} \frac{p_{ij,t}}{m_{ij,t}} \right)$$

$$= \frac{\mathcal{L}^2(f_k(\boldsymbol{x}_{i,t}; \boldsymbol{\theta}_{k,t}), y_{i,t})}{q_{ik,t}} \leq \frac{1}{q_{ik,t}}. \tag{25}$$

Taking the expectation from both sides of equation 23, we obtain

$$\sum_{t=1}^{T} \sum_{k=1}^{K} p_{ik,t} \mathcal{L}(f_k(\boldsymbol{x}_{i,t}; \boldsymbol{\theta}_{k,t}), y_{i,t}) - \sum_{t=1}^{T} \mathcal{L}(f_k(\boldsymbol{x}_{i,t}; \boldsymbol{\theta}_{k,t}), y_{i,t}) \leq \frac{\ln K}{\eta_i} + \frac{\eta_i}{2} \sum_{t=1}^{T} \sum_{k=1}^{K} \frac{p_{ik,t}}{q_{ik,t}}. \tag{26}$$

In addition, it can be written that

$$\sum_{t=1}^{T} \mathbb{E}_t[\mathcal{L}(f_{I_{i,t}}(\boldsymbol{x}_{i,t}; \boldsymbol{\theta}_{I_{i,t},t}), y_{i,t})] = \sum_{t=1}^{T} \sum_{k=1}^{K} p_{ik,t} \mathcal{L}(f_k(\boldsymbol{x}_{i,t}; \boldsymbol{\theta}_{k,t}), y_{i,t}). \tag{27}$$

Therefore, from equation 26 we arrive at

$$\sum_{t=1}^{T} \mathbb{E}_t[\mathcal{L}(f_{I_{i,t}}(\boldsymbol{x}_{i,t}; \boldsymbol{\theta}_{I_{i,t},t}), y_{i,t})] - \sum_{t=1}^{T} \mathcal{L}(f_k(\boldsymbol{x}_{i,t}; \boldsymbol{\theta}_{k,t}), y_{i,t}) \leq \frac{\ln K}{\eta_i} + \frac{\eta_i}{2} \sum_{t=1}^{T} \sum_{k=1}^{K} \frac{p_{ik,t}}{q_{ik,t}}. \tag{28}$$

According to Algorithm 1, at each learning round, client $i$ splits all models except for the chosen model into clusters. Let $\nu_{ij}$ be the minimum number of clusters when client $i$ chooses $I_{i,t} = j$ and splits all models except for model $j$ into clusters. If client $i$ employs FFD algorithm to split models, the number of clusters $m_{ij,t}$ when client $i$ chooses $I_{i,t} = j$ satisfies $m_{ij,t} \leq \frac{11}{9}\nu_{ij} + \frac{2}{3}$ (Dósa, 2007). Let $\mu_i$ be defined as $\mu_i = \max_j \nu_{ij}$. Therefore, it can be concluded that $m_{ij,t} \leq \frac{11}{9}\mu_i + \frac{2}{3} \leq 2\mu_i$. Thus, it can be written that

$$q_{ik,t} \geq p_{ik,t} + \frac{1 - p_{ik,t}}{2\mu_i} \geq \frac{1}{2\mu_i} \tag{29}$$

Combining equation 28 with equation 29 yields

$$\sum_{t=1}^{T} \mathbb{E}_t[\mathcal{L}(f_{I_{i,t}}(\boldsymbol{x}_{i,t}; \boldsymbol{\theta}_{I_{i,t},t}), y_{i,t})] - \sum_{t=1}^{T} \mathcal{L}(f_k(\boldsymbol{x}_{i,t}; \boldsymbol{\theta}_{k,t}), y_{i,t}) \leq \frac{\ln K}{\eta_i} + \eta_i \mu_i T. \tag{30}$$

which proves the lemma. $\qquad\square$

In what follows the server regret upper bound in fine-tuning model $k$ is obtained. Let $\hat{\ell}_{ik,t}$ denote the fine-tuning importance sampling loss estimate at learning round $t$ associated with the $i$-th client and the $k$-th model, defined as

$$\hat{\ell}_{ik,t} = \frac{\alpha}{q_{ik,t}} \mathcal{L}(f_k(\boldsymbol{x}_{i,t}; \boldsymbol{\theta}_{k,t}), y_{i,t}) \mathcal{I}(i \in \mathbb{G}_t, k \in \mathbb{S}_{i,t}). \tag{31}$$

According to equation 11, for any fixed $\boldsymbol{\theta}$ and $k \in [K]$, it can be written that

$$
\begin{aligned}
\|\boldsymbol{\theta}_{k,t+1} - \boldsymbol{\theta}\|^2 &= \left\| \boldsymbol{\theta}_{k,t} - \boldsymbol{\theta} - \frac{\eta_f}{N} \sum_{i=1}^{N} \nabla \hat{\ell}_{ik,t} \right\|^2 \\
&= \|\boldsymbol{\theta}_{k,t} - \boldsymbol{\theta}\|^2 - \frac{2\eta_f}{N} \sum_{i=1}^{N} \nabla^{\top} \hat{\ell}_{ik,t}(\boldsymbol{\theta}_{k,t} - \boldsymbol{\theta}) + \left\| \frac{\eta_f}{N} \sum_{i=1}^{N} \nabla \hat{\ell}_{ik,t} \right\|^2.
\end{aligned} \tag{32}
$$

Moreover, due to the convexity of the loss function $\mathcal{L}(\cdot, \cdot)$, for any learning round $t$, we find that

$$\nabla^{\top} \mathcal{L}(f_k(\boldsymbol{x}_{i,t}; \boldsymbol{\theta}_{k,t}), y_{i,t})(\boldsymbol{\theta} - \boldsymbol{\theta}_{k,t}) \leq \mathcal{L}(f_k(\boldsymbol{x}_{i,t}; \boldsymbol{\theta}), y_{i,t}) - \mathcal{L}(f_k(\boldsymbol{x}_{i,t}; \boldsymbol{\theta}_{k,t}), y_{i,t}) \tag{33}$$

Multiplying both sides of equation 33 by $\frac{\alpha \mathcal{I}(i \in \mathbb{G}_t, k \in \mathbb{S}_{i,t})}{q_{ik,t}}$, we get

$$\nabla^{\top} \hat{\ell}_{ik,t}(\boldsymbol{\theta} - \boldsymbol{\theta}_{k,t}) \leq \frac{\alpha}{q_{ik,t}} \mathcal{L}(f_k(\boldsymbol{x}_{i,t}; \boldsymbol{\theta}), y_{i,t}) \mathcal{I}(i \in \mathbb{G}_t, k \in \mathbb{S}_{i,t}) - \hat{\ell}_{ik,t}. \tag{34}$$

Summing equation 34 over clients, we obtain

$$\sum_{i=1}^{N} \hat{\ell}_{ik,t} - \sum_{i=1}^{N} \frac{\alpha}{q_{ik,t}} \mathcal{L}(f_k(\boldsymbol{x}_{i,t}; \boldsymbol{\theta}), y_{i,t}) \mathcal{I}(i \in \mathbb{G}_t, k \in \mathbb{S}_{i,t}) \leq \sum_{i=1}^{N} \nabla^{\top} \hat{\ell}_{ik,t}(\boldsymbol{\theta}_{k,t} - \boldsymbol{\theta}). \tag{35}$$

Combining equation 32 with equation 35 leads to

$$
\begin{aligned}
&\sum_{i=1}^{N} \hat{\ell}_{ik,t} - \sum_{i=1}^{N} \frac{\alpha}{q_{ik,t}} \mathcal{L}(f_k(\boldsymbol{x}_{i,t}; \boldsymbol{\theta}), y_{i,t}) \mathcal{I}(i \in \mathbb{G}_t, k \in \mathbb{S}_{i,t}) \\
&\leq \frac{N}{2\eta_f} (\|\boldsymbol{\theta}_{k,t} - \boldsymbol{\theta}\|^2 - \|\boldsymbol{\theta}_{k,t+1} - \boldsymbol{\theta}\|^2) + \frac{\eta_f}{2N} \left\| \sum_{i=1}^{N} \nabla \hat{\ell}_{ik,t} \right\|^2.
\end{aligned} \tag{36}
$$

Moreover, the expected value of $\hat{\ell}_{ik,t}$ and $\|\nabla\hat{\ell}_{ik,t}\|^2$ with respect to $\mathcal{I}(i \in \mathbb{G}_t, k \in \mathbb{S}_{i,t})$ can be obtained as

$$
\mathbb{E}_t[\hat{\ell}_{ik,t}] = \frac{\alpha}{q_{ik,t}}\mathcal{L}(f_k(\boldsymbol{x}_{i,t};\boldsymbol{\theta}_{k,t}),y_{i,t}) \times \left( \frac{p_{ik,t}}{\alpha} + \sum_{\forall j:j \neq k} \frac{p_{ij,t}}{\alpha m_{ij,t}} \right)
$$

$$
= \mathcal{L}(f_k(\boldsymbol{x}_{i,t};\boldsymbol{\theta}_{k,t}),y_{i,t}) \tag{37a}
$$

$$
\mathbb{E}_t[\|\nabla\hat{\ell}_{ik,t}\|^2] = \frac{\alpha^2}{q_{ik,t}^2}\|\nabla\mathcal{L}(f_k(\boldsymbol{x}_{i,t};\boldsymbol{\theta}_{k,t}),y_{i,t})\|^2 \times \left( \frac{p_{ik,t}}{\alpha} + \sum_{\forall j:j \neq k} \frac{p_{ij,t}}{\alpha m_{ij,t}} \right)
$$

$$
= \frac{\alpha}{q_{ik,t}}\|\nabla\mathcal{L}(f_k(\boldsymbol{x}_{i,t};\boldsymbol{\theta}_{k,t}),y_{i,t})\|^2 \leq \frac{\alpha G^2}{q_{ik,t}} \tag{37b}
$$

where the last inequality in equation 37b can be concluded from the assumption (a3) where $\|\nabla\mathcal{L}(f(\boldsymbol{x}_{i,t};\boldsymbol{\theta}_{k,t}),y_{i,t})\| \leq G$. In addition, using arithmetic mean geometric mean (AM-GM) inequality we find

$$
\left\| \sum_{i=1}^N \nabla\hat{\ell}_{ik,t} \right\|^2 \leq N \sum_{i=1}^N \|\nabla\hat{\ell}_{ik,t}\|^2. \tag{38}
$$

Therefore, using equation 37 and equation 38, taking the expectation from both sides of equation 36, it can be written that

$$
\sum_{i=1}^N \mathcal{L}(f_k(\boldsymbol{x}_{i,t};\boldsymbol{\theta}_{k,t}),y_{i,t}) - \sum_{i=1}^N \mathcal{L}(f_k(\boldsymbol{x}_{i,t};\boldsymbol{\theta}),y_{i,t})
$$

$$
\leq \frac{N}{2\eta_f}(\|\boldsymbol{\theta}_{k,t}-\boldsymbol{\theta}\|^2 - \|\boldsymbol{\theta}_{k,t+1}-\boldsymbol{\theta}\|^2) + \frac{\eta_f G^2}{2}\sum_{i=1}^N \frac{\alpha}{q_{ik,t}}. \tag{39}
$$

Summing equation 39 over learning rounds yields

$$
\sum_{i=1}^N \sum_{t=1}^T \mathcal{L}(f_k(\boldsymbol{x}_{i,t};\boldsymbol{\theta}_{k,t}),y_{i,t}) - \sum_{i=1}^N \sum_{t=1}^T \mathcal{L}(f_k(\boldsymbol{x}_{i,t};\boldsymbol{\theta}),y_{i,t})
$$

$$
\leq \frac{N}{2\eta_f}(\|\boldsymbol{\theta}_{k,1}-\boldsymbol{\theta}\|^2 - \|\boldsymbol{\theta}_{k,T+1}-\boldsymbol{\theta}\|^2) + \frac{\eta_f G^2}{2}\sum_{t=1}^T \sum_{i=1}^N \frac{\alpha}{q_{ik,t}}. \tag{40}
$$

Plugging in $\boldsymbol{\theta} = \boldsymbol{\theta}_k^*$ in equation 40 and considering the facts that $\boldsymbol{\theta}_{k,1} = \boldsymbol{0}$ and $\|\boldsymbol{\theta}_{k,T+1}-\boldsymbol{\theta}\|^2 \geq 0$, we arrive at

$$
\sum_{i=1}^N \sum_{t=1}^T \mathcal{L}(f_k(\boldsymbol{x}_{i,t};\boldsymbol{\theta}_{k,t}),y_{i,t}) - \sum_{i=1}^N \sum_{t=1}^T \mathcal{L}(f_k(\boldsymbol{x}_{i,t};\boldsymbol{\theta}_k^*),y_{i,t})
$$

$$
\leq \frac{N}{2\eta_f}\|\boldsymbol{\theta}_k^*\|^2 + \frac{\eta_f G^2}{2}\sum_{t=1}^T \sum_{i=1}^N \frac{\alpha}{q_{ik,t}} \tag{41}
$$

According to equation 29, it can be concluded that $\frac{1}{q_{ik,t}} \leq 2\mu_i$. Therefore, considering assumption (a4), we get

$$
\sum_{i=1}^N \sum_{t=1}^T \mathcal{L}(f_k(\boldsymbol{x}_{i,t};\boldsymbol{\theta}_{k,t}),y_{i,t}) - \sum_{i=1}^N \sum_{t=1}^T \mathcal{L}(f_k(\boldsymbol{x}_{i,t};\boldsymbol{\theta}_k^*),y_{i,t}) \leq \frac{NR}{2\eta_f} + \sum_{i=1}^N \mu_i \alpha \eta_f G^2 T \tag{42}
$$

which proves equation 13 and completes the proof of Theorem 1.

## C Supplementary Experimental Results and Details

The performance of both the proposed OFMS-FT method and other baseline approaches is evaluated through online image classification and online regression tasks. The image classification experiments involve the utilization of the CIFAR-10 and MNIST datasets. CIFAR-10 and MNIST are well-known computer vision datasets, comprising a total of $60,000$ and $70,000$ color images, respectively, distributed across 10 distinct classes. Each dataset includes $10,000$ test samples, with the remaining samples designated for training. To facilitate model selection, as outlined in Section 6, we train a set of 20 models using the training data from CIFAR-10 and MNIST. These models encompass two distinct architectural designs, resulting in ten models trained under each architecture. For each class label within these datasets, two models with differing architectures are trained. These models exhibit a bias towards the specific class label they are trained on, utilizing a portion of the training dataset that contains a greater number of samples from that class compared to the other classes. For the CIFAR-10 dataset, ten CNNs are trained using the VGG architecture (Simonyan & Zisserman, 2015) with 2 blocks, while the remaining ten are trained using VGG architecture with 3 blocks. The training data for each model is non-i.i.d. sampled from the $50,000$ training samples. Precisely, each CNN is trained on $9,500$ training data samples, consisting of $5,000$ samples from one class and $500$ samples drawn from the training set of each of the other nine classes. Similarly, for the MNIST dataset, ten CNNs are trained using VGG with one block, and the other ten are trained using VGG with 2 blocks. To train each model, $6,900$ data samples are drawn from the training set, with $6,000$ samples belonging to one class and $100$ samples from each of the other nine classes. Additionally, the testing data samples for CIFAR-10 and MNIST are distributed among clients in a non-i.i.d. manner. For CIFAR-10, each client receives 155 testing data samples from one class and 5 samples from each of the other nine classes. In the case of MNIST, each client receives at least 133 samples from one class and at least 5 samples from the other classes. The 200 testing data samples are randomly shuffled and are sequentially presented to each client over $T = 200$ learning rounds. Moreover, for online regression task, the performance of algorithms are tested on the following datasets (Kelly et al., 2023):

- **Air:** Each data sample has 14 features including information related to air quality such as concentration of some chemicals in the air. Data samples are collected from different geographical sites. The goal is to predict the concentration of CO in the air (Zhang et al., 2017).

- **WEC:** Each data sample has 48 features of wave energy converters. Data samples are collected from 4 different geographical sites. The goal is to predict total power output (Neshat et al., 2018).

For each regression dataset, 20 fully-connected feedforward neural networks are trained. All neural networks have 5 hidden layers each with 100 hidden neurons. ReLU activation function is employed for all hidden neurons in all networks. In order to train models for Air dataset, 10 neural networks are trained on $30,000$ samples from the site Dongsi with different initialization while other 10 neural networks are trained on $30,000$ samples of Dingling site with different initialization. In the experiments, data samples of Air dataset are distributed non-i.i.d among clients such that 50 clients observe data samples from Aotizhongxin site while other 50 clients observe data samples from Changping site. To train models for WEC dataset, 10 neural networks are trained on $70,000$ samples from the site in Sydney with different initialization. The remaining 10 neural networks are trained on $70,000$ samples from the site in Tasmania with different initialization. Data samples of WEC dataset are distributed non-i.i.d among clients such that 50 clients observe data samples from Adelaide site while other 50 clients observe data samples from Perth site. In the experiments, using Fed-OMD and PerFedAvg, each client performs one epoch of stochastic gradient descent (SGD) with learning rate of 0.001 on its batch of data to fine-tune the model. In order to perform fine-tuning, clients start to update models after 50 learning rounds so that clients can store 50 samples in batch. All experiments were carried out using Intel(R) Core(TM) i7-10510U CPU @ 1.80 GHz 2.30 GHz processor with a 64-bit Windows operating system.

Table 3 shows the average accuracy of clients along with its standard deviation on CIFAR-10 and MNIST datasets with the change in the memory budget when clients employ OFMS-FT. Moreover, Table 4 demonstrates the average MSE and its standard deviation across clients for different memory budgets on Air and

Table 3: Average and standard deviation of clients' accuracy using OFMS-FT over CIFAR-10 and MNIST with the change in budget.

| Budget | CIFAR-10 | MNIST |
|---|---|---|
| $B_i = 2, \forall i \in [N]$ | $70.01\% \pm 6.96\%$ | $89.87\% \pm 3.33\%$ |
| $B_i = 5, \forall i \in [N]$ | $76.77\% \pm 4.46\%$ | $92.05\% \pm 2.69\%$ |
| $B_i = 10, \forall i \in [N]$ | $79.90\% \pm 4.23\%$ | $92.93\% \pm 2.58\%$ |

Table 4: Average and standard deviation of clients' MSE ($\times 10^{-3}$) using OFMS-FT over Air and WEC with the change in budget.

| Budget | Air | WEC |
|---|---|---|
| $B_i = 2, \forall i \in [N]$ | $7.51 \pm 4.82$ | $8.27 \pm 1.56$ |
| $B_i = 5, \forall i \in [N]$ | $7.46 \pm 5.10$ | $7.09 \pm 1.67$ |
| $B_i = 10, \forall i \in [N]$ | $7.38 \pm 4.86$ | $6.99 \pm 1.63$ |

WEC datasets when clients use OFMS-FT. Results in Tables 3 and 4 confirm that if clients have larger memory, the accuracy of OFMS-FT improves.

We report the run times of algorithms in Table 5. Run time refers to average total run time of clients to perform their prediction task on the entire data samples that they observe up until time horizon $T$. In Table 5, OFMS-FT, $B_i = 5$, and OFMS-FT, $B_i = 2$ refer to the proposed algorithm with budgets $B_i = 5$ and $B_i = 2$, respectively. Table 5 shows that other algorithms run faster than OFMS-FT while OFMS-FT outperforms others in terms of accuracy (see Table 1). OFMS-FT runs slower since OFMS-FT evaluates and fine-tunes multiple models at each round. Comparing the run times of OFMS-FT, $B_i = 5$ with OFMS-FT, $B_i = 2$ shows that the time complexity of OFMS-FT can be controlled by budget. In time-sensitive scenarios, the budget can be chosen such that OFMS-FT can fulfill required computations before the start of the next round.

## D    Supplementary Discussions and Analysis

This section presents extended discussions on performance analysis of OFMS-FT.

### D.1    Supplementary Analysis

In sections 3 and 4, it is assumed that at each learning round $t$, each client observes one data sample and communicate with the server every learning round. This subsection analyzes the regret of OFMS-FT when clients communicate with the server every $n \geq 1$ learning rounds. Every round that clients communicate with the server called *communication round*. Therefore, the number of communication rounds $U$ is $\lfloor \frac{T}{n} \rfloor$. Let the communication $u$ occurs at learning round $\tau_u$. Without loss of generality, we can assume that $\tau_u = n(u-1) + 1$. In this case, at communication round $u$, client $i$ draws the model index $I_{i,u}$ using the PMF specified in equation 4. Then client $i$ splits all models except for model $I_{i,u}$ into clusters $\mathbb{D}_{i1,u}, \ldots, \mathbb{D}_{im_{i,u},u}$ such that the cumulative cost of models in each cluster does not exceed $B_i - c_{I_{i,u}}$. Then client $i$ draws one of the clusters uniformly at random. Let $J_{i,u}$ denote the index of the selected cluster. Client $i$ downloads all models in $J_{i,u}$-th cluster in addition to model $I_{i,u}$. Upon receiving models, client $i$ computes the importance loss estimate as

$$\ell_{ik,u} = \sum_{t=\tau_u}^{\tau_{u+1}-1} \frac{\mathcal{L}(f_k(\boldsymbol{x}_{i,t}; \boldsymbol{\theta}_{k,u}), y_{i,t})}{q_{ik,u}} \mathcal{I}(k \in \mathbb{S}_{i,u}) \tag{43}$$

Table 5: Average run time (s) of clients on CIFAR-10, MNIST, Air and WEC datasets.

| Algorithms | CIFAR-10 | MNIST | Air | WEC |
|---|---|---|---|---|
| MAB | 9.43 | 9.34 | 8.89 | 9.51 |
| Non-Fed-OMS | 57.38 | 62.09 | 57.56 | 52.70 |
| Fed-OMD | 32.80 | 23.24 | 15.86 | 15.64 |
| PerFedAvg | 47.61 | 32.21 | 19.29 | 22.03 |
| OFMS-FT, $B_i = 5, \forall i$ | 145.91 | 99.49 | 55.59 | 55.69 |
| OFMS-FT, $B_i = 2, \forall i$ | 64.42 | 43.06 | 27.78 | 28.82 |

where $\boldsymbol{\theta}_{k,u}$ denote the parameter of model $k$ between communications rounds $u$ and $u+1$ and $\mathbb{S}_{i,u}$ is a subset of models stored by client $i$ between communications rounds $u$ and $u+1$. Also, $q_{ik,u}$ can be obtained as

$$q_{ik,u} = p_{ik,\tau_u} + \sum_{\forall j : j \neq k} \frac{p_{ij,\tau_u}}{m_{ij,u}} \tag{44}$$

where $m_{ij,u}$ denote the number of model clusters at communication round $u$ if $I_{i,u} = j$. Moreover, importance sampling gradient estimate is calculated as follows by client $i$

$$\nabla \hat{\ell}_{ik,u} = \sum_{t=\tau_u}^{\tau_{u+1}-1} \frac{\alpha}{q_{ik,u}} \nabla \mathcal{L}(f_k(\boldsymbol{x}_{i,t}; \boldsymbol{\theta}_{k,u}), y_{i,t}) \mathcal{I}(i \in \mathbb{D}_u, k \in \mathbb{S}_{i,u}) \tag{45}$$

where $\mathbb{D}_u$ represents a subset of clients chosen by the server at communication round $u$ to fine-tune models. The rest of procedures and definitions are the same as Algorithm 1 and Section 3. Moreover, when clients communicate with the server every $n$ learning rounds, the $i$-th client regret $\mathcal{R}_{i,T}$ and the server regret $\mathcal{S}_{k,T}$ associated with model $k$ are defined as

$$\mathcal{R}_{i,T} = \sum_{u=1}^{U} \mathbb{E}_u \left[ \sum_{t=\tau_u}^{\tau_{u+1}-1} \mathcal{L}(f_{I_{i,u}}(\boldsymbol{x}_{i,t}; \boldsymbol{\theta}_{k,u}), y_{i,t}) \right] - \min_{k \in [K]} \sum_{u=1}^{U} \sum_{t=\tau_u}^{\tau_{u+1}-1} \mathcal{L}(f_k(\boldsymbol{x}_{i,t}; \boldsymbol{\theta}_{k,u}), y_{i,t}) \tag{46a}$$

$$\mathcal{S}_{k,T} = \frac{1}{N} \sum_{i=1}^{N} \sum_{u=1}^{U} \sum_{t=\tau_u}^{\tau_{u+1}-1} \mathcal{L}(f_k(\boldsymbol{x}_{i,t}; \boldsymbol{\theta}_{k,u}), y_{i,t}) - \frac{1}{N} \sum_{i=1}^{N} \sum_{u=1}^{U} \sum_{t=\tau_u}^{\tau_{u+1}-1} \mathcal{L}(f_k(\boldsymbol{x}_{i,t}; \boldsymbol{\theta}_k^*), y_{i,t}) \tag{46b}$$

where $\mathbb{E}_u[\cdot]$ denote the expected value given observed losses up until communication round $u$. The following Theorem obtains the regret upper bound for OFMS-FT when clients communicate with the server every $n$ learning rounds.

**Theorem 3.** *Assume that client $i$, $\forall i \in [N]$ communicates with the server every $n$ learning rounds. Under (a1) and (a2), the expected cumulative regret of the $i$-th client using OFMS-FT is bounded by*

$$\mathcal{R}_{i,T} \leq \frac{\ln K}{\eta_i} + \eta_i \mu_i nT. \tag{47}$$

*which holds for all $i \in [N]$. Under (a1)–(a4), the cumulative regret of the server in fine-tuning model $k$ using OFMS-FT is bounded by*

$$\mathcal{S}_{k,T} \leq \frac{R}{2\eta_f} + \frac{1}{N} \sum_{i=1}^{N} \eta_f \alpha \mu_i G^2 nT \tag{48}$$

*Proof.* see subSection D.2 □

If client $i$ sets

$$\eta_i = \mathcal{O}\left(\sqrt{\frac{\ln K}{\mu_i nT}}\right), \tag{49}$$

then the $i$-th client achieves sub-linear regret of

$$\mathcal{R}_{i,T} \leq \mathcal{O}\left(\sqrt{(\ln K)\mu_i nT}\right), \tag{50}$$

while the server achieves sub-linear regret of

$$\mathcal{S}_{k,T} \leq \mathcal{O}\left(\sqrt{\frac{\alpha nT}{N}\sum_{i=1}^{N}\mu_i}\right) \tag{51}$$

by setting

$$\eta_f = \mathcal{O}\left(\frac{1}{\sqrt{\frac{\alpha nT}{N}\sum_{i=1}^{N}\mu_i}}\right). \tag{52}$$

As can be inferred from theorem 3 and regret analysis presented in this subsection, the increase in $n$, degrades the regret upper bound of both clients and the server. Increase in $n$ causes that clients update their stored models fewer times and this reduces the flexibility of model selection for clients. Also, increase in $n$ leads to fine-tuning models less often which can adversely affect the prediction accuracy of models.

### D.2 Proof of Theorem 3

Substituting $\ell_{ik,t}$ with $\ell_{ik,u}$ in equation 17 and following the steps from equation 17 to equation 23, we get

$$\sum_{u=1}^{U}\sum_{k=1}^{K}p_{ik,\tau_u}\ell_{ik,u} - \sum_{u=1}^{U}\ell_{ik,u} \leq \frac{\ln K}{\eta_i} + \frac{\eta_i}{2}\sum_{u=1}^{U}\sum_{k=1}^{K}p_{ik,\tau_u}\ell_{ik,u}^2. \tag{53}$$

Moreover, the expected value of $\ell_{ik,u}$ and $\ell_{ik,u}^2$ given observed losses till communication round $u$, can be obtained as

$$\mathbb{E}_u[\ell_{ik,u}] = \left(\sum_{t=\tau_u}^{\tau_{u+1}-1}\frac{\mathcal{L}(f_k(\boldsymbol{x}_{i,t};\boldsymbol{\theta}_{k,u}),y_{i,t})}{q_{ik,u}}\right) \times \left(p_{ik,\tau_u} + \sum_{\forall j:j\neq k}\frac{p_{ij,\tau_u}}{m_{ij,u}}\right)$$

$$= \sum_{t=\tau_u}^{\tau_{u+1}-1}\mathcal{L}(f_k(\boldsymbol{x}_{i,t};\boldsymbol{\theta}_{k,u}),y_{i,t}). \tag{54}$$

Furthermore, using arithmetic-mean geometric-mean (AM-GM) inequality, $\ell_{ik,u}^2$ can bounded from above as

$$\ell_{ik,u}^2 \leq n\left(\sum_{t=\tau_u}^{\tau_{u+1}-1}\left(\frac{\mathcal{L}(f_k(\boldsymbol{x}_{i,t};\boldsymbol{\theta}_{k,u}),y_{i,t})}{q_{ik,u}}\mathcal{I}(k\in\mathbb{S}_{i,u})\right)^2\right). \tag{55}$$

Moreover, based on the assumption that $0 \leq \mathcal{L}(f_k(\boldsymbol{x}_{i,t};\boldsymbol{\theta}_{k,u}),y_{i,t}) \leq 1$, given the observed losses in prior rounds, expected value of $\ell_{ik,u}^2$ can be bounded from above as

$$\mathbb{E}_u\left[\left(\frac{\mathcal{L}(f_k(\boldsymbol{x}_{i,t};\boldsymbol{\theta}_{k,u}),y_{i,t})}{q_{ik,u}}\mathcal{I}(k\in\mathbb{S}_{i,u})\right)^2\right] = \frac{\mathcal{L}^2(f_k(\boldsymbol{x}_{i,t};\boldsymbol{\theta}_{k,u}),y_{i,t})}{q_{ik,u}^2} \times \left(p_{ik,\tau_u} + \sum_{\forall j:j\neq k}\frac{p_{ij,\tau_u}}{m_{ij,u}}\right)$$

$$= \frac{\mathcal{L}^2(f_k(\boldsymbol{x}_{i,t};\boldsymbol{\theta}_{k,u}),y_{i,t})}{q_{ik,u}} \leq \frac{1}{q_{ik,u}}. \tag{56}$$

Combining equation 55 with equation 56, we arrive at

$$\mathbb{E}_u[\ell_{ik,u}^2] \leq \frac{n^2}{q_{ik,u}}. \tag{57}$$

Taking the expectation from both sides of equation 53 and considering the fact that $\xi_i \geq 0$, it can be concluded that

$$\sum_{u=1}^{U}\sum_{t=\tau_u}^{\tau_{u+1}}\sum_{k=1}^{K}p_{ik,\tau_u}\mathcal{L}(f_k(\boldsymbol{x}_{i,t};\boldsymbol{\theta}_{k,u}),y_{i,t}) - \sum_{u=1}^{U}\sum_{t=\tau_u}^{\tau_{u+1}}\mathcal{L}(f_k(\boldsymbol{x}_{i,t};\boldsymbol{\theta}_{k,u}),y_{i,t})$$

$$\leq \frac{\ln K}{\eta_i} + \frac{\eta_i n^2}{2}\sum_{u=1}^{U}\sum_{k=1}^{K}\frac{p_{ik,\tau_u}}{q_{ik,u}}. \tag{58}$$

Moreover, it can be written that

$$\mathbb{E}_u\left[\sum_{t=\tau_u}^{\tau_{u+1}-1}\mathcal{L}(f_{I_{i,u}}(\boldsymbol{x}_{i,t};\boldsymbol{\theta}_{k,u}),y_{i,t})\right] = \sum_{t=\tau_u}^{\tau_{u+1}}\sum_{k=1}^{K}p_{ik,\tau_u}\mathcal{L}(f_k(\boldsymbol{x}_{i,t};\boldsymbol{\theta}_{k,u}),y_{i,t}). \tag{59}$$

Considering the facts that equation 29 holds true for $p_{ik,\tau_u}$ and $q_{ik,u}$, $\forall k \in [K]$ and $nU = T$, we can conclude that

$$\sum_{u=1}^{U}\mathbb{E}_u\left[\sum_{t=\tau_u}^{\tau_{u+1}-1}\mathcal{L}(f_{I_{i,u}}(\boldsymbol{x}_{i,t};\boldsymbol{\theta}_{k,u}),y_{i,t})\right] - \sum_{u=1}^{U}\sum_{t=\tau_u}^{\tau_{u+1}}\mathcal{L}(f_k(\boldsymbol{x}_{i,t};\boldsymbol{\theta}_{k,u}),y_{i,t})$$

$$\leq \frac{\ln K}{\eta_i} + \eta_i\mu_i nT \tag{60}$$

which obtains the regret of client $i$ using OFMS-FT when the $i$-th client communicates with the server every $n$ learning rounds. Similar to $\hat{\ell}_{ik,t}$, define $\hat{\ell}_{ik,u}$ as

$$\hat{\ell}_{ik,u} = \sum_{t=\tau_u}^{\tau_{u+1}-1}\frac{\alpha}{q_{ik,u}}\mathcal{L}(f_k(\boldsymbol{x}_{i,t};\boldsymbol{\theta}_{k,u}),y_{i,t})\mathcal{I}(i \in \mathbb{D}_u, k \in \mathbb{S}_{i,u}) \tag{61}$$

Moreover, substituting $\hat{\ell}_{ik,t}$ with $\hat{\ell}_{ik,u}$ in equation 32 and following the derivation steps from equation 32 to equation 36, we obtain

$$\sum_{i=1}^{N}\hat{\ell}_{ik,u} - \sum_{i=1}^{N}\sum_{t=\tau_u}^{\tau_{u+1}-1}\frac{\alpha}{q_{ik,u}}\mathcal{L}(f_k(\boldsymbol{x}_{i,t};\boldsymbol{\theta}),y_{i,t})\mathcal{I}(i \in \mathbb{D}_u, k \in \mathbb{S}_{i,u})$$

$$\leq \frac{N}{2\eta_f}(\|\boldsymbol{\theta}_{k,u} - \boldsymbol{\theta}\|^2 - \|\boldsymbol{\theta}_{k,u+1} - \boldsymbol{\theta}\|^2) + \frac{\eta_f}{2N}\left\|\sum_{i=1}^{N}\nabla\hat{\ell}_{ik,u}\right\|^2. \tag{62}$$

Expected value of $\hat{\ell}_{ik,u}$ and $\|\nabla\hat{\ell}_{ik,u}\|^2$ can be obtained as

$$\mathbb{E}_u[\hat{\ell}_{ik,u}] = \sum_{t=\tau_u}^{\tau_{u+1}-1}\frac{\alpha}{q_{ik,u}}\mathcal{L}(f_k(\boldsymbol{x}_{i,t};\boldsymbol{\theta}_{k,u}),y_{i,t}) \times \left(\frac{p_{ik,\tau_u}}{\alpha} + \sum_{\forall j:j\neq k}\frac{p_{ij,\tau_u}}{\alpha m_{ij,u}}\right)$$

$$= \sum_{t=\tau_u}^{\tau_{u+1}-1}\mathcal{L}(f_k(\boldsymbol{x}_{i,t};\boldsymbol{\theta}_{k,u}),y_{i,t}) \tag{63a}$$

$$\mathbb{E}_u[\|\nabla\hat{\ell}_{ik,u}\|^2] = \frac{\alpha^2}{q_{ik,u}^2}\left\|\sum_{t=\tau_u}^{\tau_{u+1}-1}\nabla\mathcal{L}(f_k(\boldsymbol{x}_{i,t};\boldsymbol{\theta}_{k,u}),y_{i,t})\right\|^2 \times \left(\frac{p_{ik,\tau_u}}{\alpha} + \sum_{\forall j:j\neq k}\frac{p_{ij,\tau_u}}{\alpha m_{ij,u}}\right)$$

$$= \frac{\alpha}{q_{ik,u}}\left\|\sum_{t=\tau_u}^{\tau_{u+1}-1}\nabla\mathcal{L}(f_k(\boldsymbol{x}_{i,t};\boldsymbol{\theta}_{k,u}),y_{i,t})\right\|^2$$

$$\leq \frac{\alpha n}{q_{ik,u}}\sum_{t=\tau_u}^{\tau_{u+1}-1}\|\nabla\mathcal{L}(f_k(\boldsymbol{x}_{i,t};\boldsymbol{\theta}_{k,u}),y_{i,t})\|^2 \leq \frac{\alpha n^2 G^2}{q_{ik,u}} \tag{63b}$$

where the last two inequalities in equation 63b obtained using AM-GM inequality and the assumption that $\|\nabla \mathcal{L}(f_k(\boldsymbol{x}_{i,t}; \boldsymbol{\theta}_{k,u}), y_{i,t})\|^2 \leq G^2$. Moreover, using AM-GM inequality and equation 63b, we can write that

$$\mathbb{E}_u \left[ \left\| \sum_{i=1}^{N} \nabla \hat{\ell}_{ik,u} \right\|^2 \right] \leq N \sum_{i=1}^{N} \mathbb{E}_u[\|\nabla \hat{\ell}_{ik,u}\|^2] \leq N \sum_{i=1}^{N} \frac{\alpha n^2 G^2}{q_{ik,u}}. \tag{64}$$

Taking the expectation from both sides of equation 62, we get

$$\sum_{i=1}^{N} \sum_{t=\tau_u}^{\tau_{u+1}-1} \mathcal{L}(f_k(\boldsymbol{x}_{i,t}; \boldsymbol{\theta}_{k,u}), y_{i,t}) - \sum_{i=1}^{N} \sum_{t=\tau_u}^{\tau_{u+1}-1} \mathcal{L}(f_k(\boldsymbol{x}_{i,t}; \boldsymbol{\theta}), y_{i,t})$$
$$\leq \frac{N}{2\eta_f}(\|\boldsymbol{\theta}_{k,u} - \boldsymbol{\theta}\|^2 - \|\boldsymbol{\theta}_{k,u+1} - \boldsymbol{\theta}\|^2) + \frac{\eta_f}{2} \sum_{i=1}^{N} \frac{\alpha n^2 G^2}{q_{ik,u}}. \tag{65}$$

Following derivation steps from equation 39 to equation 41, using equation 65 we can obtain

$$\sum_{i=1}^{N} \sum_{u=1}^{U} \sum_{t=\tau_u}^{\tau_{u+1}-1} \mathcal{L}(f_k(\boldsymbol{x}_{i,t}; \boldsymbol{\theta}_{k,u}), y_{i,t}) - \sum_{i=1}^{N} \sum_{u=1}^{U} \sum_{t=\tau_u}^{\tau_{u+1}-1} \mathcal{L}(f_k(\boldsymbol{x}_{i,t}; \boldsymbol{\theta}_k^*), y_{i,t})$$
$$\leq \frac{N\|\boldsymbol{\theta}_k^*\|^2}{2\eta_f} + \frac{\eta_f}{2} \sum_{i=1}^{N} \sum_{u=1}^{U} \frac{\alpha n^2 G^2}{q_{ik,u}}. \tag{66}$$

Considering the fact that $q_{ik,u} \geq \frac{1}{2\mu_i}$ (see equation 29), using equation 66 we arrive at

$$\sum_{i=1}^{N} \sum_{u=1}^{U} \sum_{t=\tau_u}^{\tau_{u+1}-1} \mathcal{L}(f_k(\boldsymbol{x}_{i,t}; \boldsymbol{\theta}_{k,u}), y_{i,t}) - \sum_{i=1}^{N} \sum_{u=1}^{U} \sum_{t=\tau_u}^{\tau_{u+1}-1} \mathcal{L}(f_k(\boldsymbol{x}_{i,t}; \boldsymbol{\theta}_k^*), y_{i,t})$$
$$\leq \frac{N\|\boldsymbol{\theta}_k^*\|^2}{2\eta_f} + \sum_{i=1}^{N} \eta_f \alpha \mu_i G^2 nT \tag{67}$$

which proves the theorem.

