# OpenReview forum: "Budgeted Online Model Selection and Fine-Tuning via Federated Learning"
_TMLR — Accepted by TMLR_

### Review · Reviewer_CpQM · 2023-11-03

**Summary Of Contributions:**

The paper focuses on Online Federated Learning (OFL). The contributions are threefold: firstly, it proposes leveraging Online Model Selection (OMS) to achieve better client-side accuracy in OFL. Secondly, it designs an algorithm to select models in a scenario where the client has limited memory and cannot transfer all models from the server to make OMS locally. Finally, it proposes a method to deal with limited bandwidth on the server side for incoming client model updates.

The authors support their algorithm design with a detailed regret analysis. Furthermore, they design experiments using classification and regression tasks. They adapt several baseline algorithms to this setting and show their method is competitive compared to them.

I would like to thank the authors for sharing their research as I believe this paper has contributions that are in TMLR's audience's interest.

**Audience:**

Yes

**Broader Impact Concerns:**

I do not have concerns about Broader Impact.

**Claims And Evidence:**

No

**Requested Changes:**

Critical:

1. In simulated FL the data distribution is crucial for the model performance. It should be discussed in the main paper, not in the appendix.
2. What motivates the choice of 1 majority class in the data distribution? 2 majority classes can be also seen in non-IID experiments in the FL literature. Is it typical in OFL?
3. It is not clear to me in the experiment description:
    a) does MAB use the same pre-trained model as Fed-OMD, PerFedAvg, and OFMS-FT at the start?
    b) Does Non-Fed-OMS train the locally stored models?
4. in case Non-Fed-OMS does not do this, I am missing a baseline where clients and server train $B$ models together, while clients select one of them using OMS locally. This would support the use of more server-stored models.
5. I am missing a baseline where the server stores the same number of models as OFMS-FT, but the clients select the $B_i$ models randomly. This would support the use of the defined $z$ weights.

If server-side client selection is in the scope of the paper:

6. Server-side client selection is not compared to any baseline (or not discussed in section 6's last paragraph) Either say it is out of the scope of the paper or support with more experiments.
7. In the current regression tasks, there are 20 models, 100 clients, clients update $B=5$ models, and the update bandwidth $\Omega=250$ ("only half the clients are able to send their updated model to the server"). Because the clients have very unbalanced datasets, the server might benefit more from seeing all the clients in all rounds resulting in an overall balanced set of data in the training rounds. E.g. this can be achieved if the clients reduce their model memory to half, each client updates fewer models but the server sees all clients (with the current setup it would mean $2.5" models from each client.
8. Client selection is missing from the related works See this taxonomy as a reference: [r1]
9. Server-side client selection should be evaluated compared to SoTA client selections. For example [r2] also works with gradient estimates for unseen clients.

Minor:

10. Section 2. "Each model..." sentence defines $\mathcal{X}$ and $\mathcal{Y}$ but no part of the paper builds on this sentence, which could be removed for a more clear read.
11. Section 3.1 PMF acronym never defined.
12. $\mathcal{F}$ and $\mathbb{F}$ font styles are both representing sets, while $\mathcal{F}$ also represents vector spaces and probabilities. It could be confusing.
13. The number of model clusters is represented with $m$ (in section 3.1 $m_{i,t}$ for the $i$ client in $t$ round and $m$* for the minimum), $o_{ij,t}$ in the same section a few paragraph later and $v_{i,j}$ for the minimum number of clusters in section 4.
14. At the end of section 3.1 I would add clarity on the exp() function's behavior: it keeps the weights the same if not updated. Otherwise, the higher the loss the more the weight is reduced.
15. An experiment where more small models than big ones can fit into the clients might be interesting. With $c_{small}=0.89$ and $B=5$ it is the same 5 models per clients as $c_{big}=1$


[r1] Németh, G. D., Lozano, M. A., Quadrianto, N., & Oliver, N. M. (2022). A Snapshot of the Frontiers of Client Selection in Federated Learning. Transactions on Machine Learning Research.

[r2] Chen, T., Giannakis, G., Sun, T., & Yin, W. (2018). LAG: Lazily aggregated gradient for communication-efficient distributed learning. Advances in neural information processing systems, 31.

I think overall the paper has clear contributions and by fixing 1-5 it could be strengthened to an even better level. 6-9 are only critical if the client selection is in the scope of the paper. 10-15 would improve the clarity of the paper in my opinion.

**Strengths And Weaknesses:**

Strengths:
 - The paper introduces and solves a relevant OFL problem.
 - The motivation of the algorithm design is well supported with a mathematical theorem.
 - The theoretic effectiveness is supported by a thorough regret analysis.

Weaknesses:
 - Some experiment design choices are not motivated.
 - The baseline algorithms are not adapted to the experiment setup in a way that allows fair comparison.
 - The 3rd contribution, server-side client selection is not compared to SoTA client selection in federated learning, and the related works section does not discuss this line of research.

---

> ### Author Response · Authors · 2023-12-02
> **Response to Comments**
>
> We would like to express our gratitude for taking the time to review our submission and letting us know your thoughtful comments. In response to your comments, we add more baselines to our experiments, expand related works discussion and improve the presentation of the paper. We highlight our revisions in blue. Please find below our responses to your comments.
>
> ## Experiments
> 1. We add more explanations about the data distribution to the main text of the paper in the revised version. This revision can be found on pages 10 and 11 in section 6 highlighted in blue.
> 2. In image classification experiments designed to assess the proposed algorithm's performance on non-i.i.d data distributions, we distribute data samples heterogeneously based on labels. Each client is exposed to data samples associated with a single majority class label, creating a scenario that explores the algorithm's ability to handle heterogeneous data. This experimental setup, while hypothetical, allows us to evaluate algorithmic performance under diverse data conditions. Various alternative setups, such as observing two majority classes, can also be considered. Given the fact that in experiments clients do not have any information about data distribution, the proposed algorithm performance in comparison to other baselines remains robust across varying data distributions. Additionally, to account for different types of heterogeneity in data distribution among clients, our regression experiments involve distributing data non-i.i.d based on data features rather than labels.
> 3. MAB uses the same pre-trained models as Fed-OMD, PerFedAvg and OFMS-FT at the start. And Non-Fed-OMS does not locally fine-tune models.
> 4. Based on your comments, we add a baseline to this revised version called B-Fed-OMFT. B-Fed-OMFT stands for Budgeted Federated Online Model Fine-Tuning. In this approach, the server maintains a set of models that can be fit into the memory of all clients. Clients collaborate with the server to fine-tune all models in each learning round. In the B-Fed-OMFT framework, each client employs the Exp3 algorithm to choose one model to perform the prediction task. The experimental results show that OFMS-FT outperformed B-Fed-OMFT which supports using a large dictionary of models.
> 5. Based on your comments, we add another baseline to Table 1 which is called RMS-FT. RMS-FT denote a baseline where at each learning round each client chooses a subset of models uniformly at random to fine-tune them. The prediction task is then carried out by selecting one of the chosen models uniformly at random. As can be seen from Table 1, OFMS-FT outperformed RMS-FT and this shows the effectiveness of using $z$ weights.
>
> ## Client Selection
> The client selection is out of the scope of this paper. The client selection in online federated learning is quite an unexplored research area which is different from online model selection and fine-tuning. To clarify this in the paper, we add a discussion to the related works section to discuss client selection. This revision is highlighted in blue and can be found on page 9.
>
> Based on the theoretical results reducing the model memory to half and letting all clients to send their updates, results in looser regret bounds compared to the current regression setting. Reducing the memory of clients to half reduces $\mu_i$ by half approximately. Conversely, this reduction in client memory doubles the value of $\alpha$. Consequently, in light of the regret bounds outlined in equations (14) and (15), halving client memory does not enhance server regret but increases client regret by a factor of $\sqrt{2}$. The reason behind this is that halving the client memory results in halving client loss observations. This degrades the performance of clients in model selection. However, doubling the chance of being chosen by the server to send updates while the memory is reduced to half does not change the chance that client $i$ participates in fine-tuning model $k$ at learning round $t$. Therefore, theoretically reducing the client memory to half and doubling the number of clients that can send their updates does not improve server regret.
>
> ## Minor Comments
> * In order to improve the readability of the paper, we omit the notations $\mathcal X$ and $\mathcal Y$.
> * We define the PMF acronym in this revised version.
> * In this revised version all sets are represented by $\mathbb F$ to avoid any confusion.
> * To make notations more consistent we replace both the notations $o_{ij,t}$ and $m_{i,t}$ with one notation $m_{ij,t}$. Thus, $m_{ij,t}$ denote the number of clusters constructed by client $i$ at learning round $t$ if $I_{i,t}=j$. Also instead of $\nu_{i,j}$ we use the notation $m_{ij}^*$ in section 4. Hence in this revised version $m_{ij}^*$ denote the minimum number of clusters if client $i$ splits all models except for model $j$ such that the cumulative cost of each cluster does not exceed $B_i - c_j$.

---

> > ### Comment · Reviewer_CpQM · 2023-12-05
> >
> > Thank you for the update. They addressed my concerns and I recommended the paper for acceptance.

---

### Review · Reviewer_qwhj · 2023-11-17

**Summary Of Contributions:**

The paper studes the problem of online model selection in a federated learning scenario in which the communication and the storage space of the clients are limited. The authors proposed a novel algorithm based on the clustering of the available models and their update according to the losses. They study the bounds on the regret from an adversarial point of view and provide some experiments to validate the proposed algorithm.

**Audience:**

Yes

**Broader Impact Concerns:**

I do not foresee any ethical concern being a theoretical work

**Claims And Evidence:**

Yes

**Requested Changes:**

Comments and minor changes:
"privately determined by the environment" it is not clear why
i.e. -> i.e., same for e.g.
The problem formulation lacks of the definition of the objective that algorithms has to optimize.
(5) -> Equation (5) similarly in the entire paper
Define the acronyms
Add punctuation to formulas
section 4 -> Section 4
"bin-packing algorithms such as FFD can be employe" please provide a reference
Please anticipate the definition of the regret, for instance in the problem formulation
Did you evaluate the errors and accuracies over a test set for each client and then averaged?

**Strengths And Weaknesses:**

I think that the motivation for the setting is strong, however, the assumption that the losses are adversarial is not realistic to me. I think that a more suitable modeling would either assume a stochastic stationary setting, in which the dataset we are using for the online learning processs are fixed, or a non.stationsary setting, in which the dataset presents a variation over time for stochastic reasons.

Following the description of the algorithm is difficult. I would suggest providing the description with the corresponding lines of the pseudocode.
Moreover, adding an overall description of the algorithm could be beneficial for its comprehension.

The experimental part is convincing. I would have also appreciated a more detailed sensitivity analysis, e.g., on the budget parameters. For instance, how is the performance when the budgets for the different clients are different?

---

> ### Author Response · Authors · 2023-12-03
> **Response to Comments**
>
> Thank you very much for reviewing and manuscript and letting us know your valuable comments. In response to your comments, we add more experimental results to the paper and more explanations about the setting studied by this paper. Our revisions are highlighted in blue. Please find below our responses to your comments.
>
> * We would like to clarify that in our setting the environment does not determine the loss adversarially. Instead the environment determines the label through a process unknown to clients and the label is not necessarily chosen to increase the loss. In fact, as it is pointed out by the last two sentences in the first paragraph of section 2, this paper considers the non-stationary cases where data distribution changes over time while the distribution is unknown. In online learning literature, usually the settings are divided into either stochastic or adversarial. Usually stochastic setting refers to cases where data distribution is stationary. Therefore, we believe that our setting is more similar to the adversarial case. However, the adversarial setting studied by this paper is different from some adversarial settings in the literature. To make this clear in the revised version, we state that the environment chooses the label through a process unknown to clients and also the last 5 lines of the first paragraph of section 3 explain the setting studied by this paper. The revision is highlighted in blue.
> * We add a description of Algorithm 1 in the last paragraph of section 3 on page 5. This revision is highlighted in blue.
> * We add more results on the sensitivity analysis on the budget. Specifically, we add Table 2 to the paper which presents the sensitivity of the MSE and its standard deviation, as achieved by the proposed OFMS-FT, to the budget $B_i$ ($\forall i \in [N]$) over the WEC dataset. In this configuration, the budget varies across clients, with a subset having $B_i=3$ and the remainder $B_i=5$. The improvement in MSE becomes evident as the number of clients with a budget of $B_i=5$ increases. This revision can be found on page 11 highlighted in blue.
>
> ## Minor Changes
> * We revised “privately determined by the environment” to “is determined by the environment through a process unknown to clients”. This revision can be found on page 2 highlighted in blue.
> * We fixed the mentioned issue with i.e. and e.g. in the entire paper.
> * To address your concern, we move the definition of clients regret and the server regret to section 2. We clarify that the objective of clients and the server is to minimize their regret. This revision can be found on pages 2 and 3.
> * In order to refer to equations in this revised version, we use “equation 5” instead of “(5)”.
> * We define the acronym PMF in this revised version.
> * We add punctuations to formulas in the revised version.
> * We revised “section 4” to “Section 4” and we fixed the similar issue in the entire paper.
> * In this revised version, we add reference for FFD in section 3.2. Please note that in the initial submission we cited the FFD paper in section 3.1.
> * We move the definition of regret from section 4 to section 2 based on your comment.
> * The results reported in the paper are accuracies and MSE average across clients. This means we obtain the accuracy achieved by each client on its observed data samples and then we report the average performance as well as its standard deviation across clients.

---

### Review · Reviewer_EDLF · 2023-11-18

**Summary Of Contributions:**

This paper considers the problem of online model selection - where data arrives sequentially and an agent wishes to select an appropriate prediction model from a set ('dictionary') of candidate models. The particular novelty in this paper is to consider a budgeted setting, where an agent has constraints on how many candidate models it can consider. This is motivated by learning on devices with limited storage.

This broad challenge is tackled in a specific setting where a group of learners ('clients'), who have limited storage capacity, learn in a federated manner governed by a server, which has sufficient capacity to store all models in the dictionary. Clients store (time-varying) subsets of models from the dictionary and send updated parameters (the 'fine-tuning' step referenced in the title) to the server following rounds of prediction.

The contribution of the paper is an algorithm for clients to employ in this regime (OFMS-FT), an analysis of the regret of OFMS-FT (which yields a sub-linear upper bound in an adversarial setting with constraints on the amount of communication between clients and the server), and experimental results which compare OFMS-FT to other algorithms which are not designed for the present task but for online federated learning (without model selection), or online model selection (without federation), or federated model selection (but offline). OFMS-FT outperforms these in terms of cumulative prediction accuracy metrics.

**Audience:**

Yes

**Broader Impact Concerns:**

This isn't explicitly so far as I can see. I would consider it good practice to add a reflection on this in the Appendix, but I don't think it is critical for this paper, as the work is fairly far from applications and I see any discussion of negative impacts as likely being highly speculative, as is typical for this flavour of theoretical/methodological online learning papers.

**Claims And Evidence:**

Yes

**Requested Changes:**

My requested changes follow from the weakness mentioned above. The most critical is to address the point about server's regret, but I would also consider clarity on the competitor algorithms and questions about the experiments critical to securing an acceptance recommendation. The remainder are just points to consider to strengthen, but I'd be disappointed if the bullet pointed minor remarks couldn't be addressed as I think they are straightforward and not contentious.

**Strengths And Weaknesses:**

The strengths of the paper are 1. That the problem is fairly interesting -  model selection in online learning is a popular and important topic, and methods which are aware of storage and computational budgets are likely to become more and more popular as models continue to become larger, and sustainability/efficiency becomes increasingly important in ML. 2. That the problem is well specified and the contribution of the paper is well-defined, and (as far as I am aware) novel. 3. That the proposed method is effective on the provided experiments, outperforming some reasonable competitors.

My biggest concern is around the definition of (and subsequent results on) the server’s regret (12). I think it is possible for this term to become negative for sufficiently large $T$, since the performance of the best possible single action in hindsight could be beaten by the performance of actions which are optimised on a per-client basis, which I believe the OFMS-FT algorithm allows for. As such, it’s not obvious how good $\tilde{O}(\sqrt{T})$ regret is for this problem, without further lower bound analysis of this quantity. Curiously $S_T$ is actually higher than $\sum_{i=1}^N {R}_{i,T}/N$, the opposite of what I would have expected. I would like to be convinced either that I am mistaken in my concerns about ${S}_T$, that something can be added to justify the usefulness of the ${S}_T$ result, or that something can be added to justify how difficult it would be to go further than this and that what is presented is state-of-the-art and raises an interesting open question. To be sure, I believe that while the section ‘Challenges of Obtaining Regret in (16)’ does consider the challenges induced by communication constraints, it doesn’t address the fact that there may exist a combination of models per agent that may have lower loss in hindsight than the best model overall in hindsight. This is an essential change for me, and currently is preventing me giving a positive score. My subsequent paragraphs of comments are ordered in decreasing importance.

I’m not totally clear on how Fed-OMD is being used – is it that a single model is picked rather than the dictionary, or a subset? If so, which model(s) (is it at random)? Similarly, I don’t think the nature of PedFedAvg is clear to the reader from the short comments made about it, and I was curious why it is not mentioned in the main related work section if it is related enough to run in the experiments? Section 5 should at least reference the existence of the material in Section D.

Two questions re the experiments: 1. I found it hard to figure out how many meta-replications the experimental results are averaged over – can you clarify that, and 2. Did you do experiments with different learning rates? If not, how would you recommend someone applying this model chooses the learning rate for a new problem?

I was curious as to whether you think the selection of clients for parameter fine-tuning could be done better than uniformly at random. Would it have benefitted the server to know which clients have larger/smaller gradients and adjust for this somehow? Or is it expected that the randomised choice of $\mathbb{C}_t$ will work as well as can be achieved given the budget $\Omega$?

There are parts where the notation could be more intuitive. For instance, decision variables could all be the same kind of symbol, but there are some which are upper case Latin letters, some are lower case and some are Greek. Similarly it is often intuitive to keep similar letters for similar purposes – this is sometimes done k indexes the set {1,…,K} but $b_k$’s sum to $\beta$ which is bounded by $\Omega$, while $B$ is being used to bound a different notion of cost for instance; elsewhere $c$ is a cost, $\mathcal{C}$ is a cluster, $\mathbb{C}$ is a group of clients, and $C$ is a bound on the norm of regression parameters. There is a lot going on notationally and algorithmically in the paper and I feel the reader has to work harder to follow it because of some of these choices of notation.


Some points that are minor, but I think easy to fix and worth raising:
-	Section 1 Paragraph 1: ‘the sheer volume of data’ is a weird phrasing here, maybe just ‘the data’ would be fine.
-	Section 2: I don’t think $T$ is ever formally defined.
-	Section 2 Paragraph 1: ‘larger storage capacity store models’ should be ‘larger storage capacity stores models’
-	Section 2 Para 1: ‘Let [K] denotes the set’ should be ‘Let [K] denote the set’ (similar on page 3 and with other verbs in multiple parts of the manuscript)
-	Section 2 Para 1: ‘studies adversarial setting’ should be ‘studies the adversarial setting’ (again occurs in multiple instances)
-	Section 2 Para 1: should it be that the environment chooses the label rather than the loss adversarially? If it chooses loss, and different labels could have the same loss, then you need to assume another rule for how losses map to labels; or explicitly assume a 1-1 mapping from loss to label for all x, theta, and I. If you just let the environment choose the label, you shouldn’t need to worry about this.
-	Section 3.1 Para 1: there are grammar issues in the last sentence
-	Section 3 and onwards: I think there are a lot of equations numbered that are not referenced?
-	Section 4: Should (a1) also be an assumption on all $k \in [K]$?
-	References: Cella et al. and Deng et al. do not have a publication venue associated with the reference.

---

> ### Author Response · Authors · 2023-12-03
> **Server Regret and Experiments**
>
> We would like to express our gratitude for your review and letting us know your valuable comments. In response to your comments, we revise the definition of server regret, add more discussions to the paper and improve the presentation. Our revisions are highlighted in blue. Please find below our point-by-point response to your comments.
>
> ## Server Regret
> Based on this comment, we revise the definition of server regret in the paper. The new definition can be found in equation (3) on page 3. Based on the comments of other reviewers, we move the definition of regret from section 4 to section 2 in this revised version. The revised definition of the server regret in fine-tuning model $k$ is the difference between the loss of fine-tuned model $k$ and the loss of model $k$ with the optimal parameter. According to steps 5-10 of the Algorithm 1, using the proposed OFMS-FT, the server is not involved in local model selection performed by clients. The server only collaborates with clients to fine-tune models (please see steps 12-14). The previous definition of the server regret involves the local model selection while the server does not have any control on local model selection. This led to the problem pointed out in your review. The revised definition of the server regret measures the performance of the server in fine-tuning each model independent of local model selection performed by each client. The revised definition of server regret examines how good the fine-tuned model performs compared to the best in hindsight and server regret cannot be negative.
>
> ## Experiments
> * In Fed-OMD, clients and the server collaborate to fine-tune a single model. All clients fine-tune and perform the prediction with the same model and there is not any model selection in Fed-OMD. We chose one of the models randomly for Fed-OMD and clients collaborated to fine-tune that model. Note that Fed-OMD originally has been proposed for online federated learning with one single model. We add this baseline to compare the performance of online federated model selection and fine-tuning with online federated learning with a single model. To make this clear on page 10 we state that “Higher accuracy of OFMS-FT compared with Fed-OMD and PerFedAvg indicates the benefit of fine-tuning multiple models rather than one.” Furthermore, to address your concern about the relevance of PerFedAvg, we move the related works discussions in the appendix to the main related work section in the paper. This revision is highlighted in blue in section 5.
>
> * Moreover, we add a new baseline to this revised version called B-Fed-OMFT. B-Fed-OMFT stands for Budgeted Federated Online Model Fine-Tuning. In this approach, the server maintains a set of models that can be fit into the memory of all clients. Clients collaborate with the server to fine-tune all models in each learning round. In the B-Fed-OMFT framework, each client employs the Exp3 algorithm to choose one model to perform the prediction task. The experimental results show that OFMS-FT outperformed B-Fed-OMFT which supports using a large dictionary of models.
>
> * We perform the experiments with a single replication maintaining consistency by using the same random seed for both the proposed OFMS-FT and all baseline methods. Performing experiments on multiple meta-replications over all datasets and baselines is time-consuming in our case. Furthermore, we performed the experiments on CIFAR-10 for 3 meta-replications and we observed minimal variations in the performance of the proposed OFMS-FT and other baselines compared to the reported results in Table 1. For example, the accuracy of the proposed OFMS-FT on 3 meta-replications is 76.51% $\pm$ 4.68%. We explain that we perform experiments with a single replication on page 9 in the section 6 with relevant revisions highlighted in blue.
>
> * We employed learning rates $\eta_i = 10/\sqrt{T}$ and $\eta_f = 10^{-3}/\sqrt{T}$. As it is discussed on pages 6 and 7, the theoretical results suggest that both $\eta_i$ and $\eta_f$ should be of order $\mathcal O(1/\sqrt{T})$. Thus, the learning rates $\eta_i$ and $\eta_f$ can be expressed in the form $a_1/\sqrt{T}$ and $a_2/\sqrt{T}$, where $a_1$ and $a_2$ are experimentally determined constants. We performed a few experiments on CIFAR-10 and we observed that $\eta_i = 10/\sqrt{T}$ and $\eta_f = 10^{-3}/\sqrt{T}$ obtained good performance. Consequently, we have adopted these learning rates for all other datasets in our experiments. Additionally, based on theoretical considerations and our empirical findings, we recommend utilizing $\eta_i = 10/\sqrt{T}$ and $\eta_f = 10^{-3}/\sqrt{T}$ as default learning rates for new problems.

---

> ### Author Response · Authors · 2023-12-03
> **Client Selection, Notation and Minor Comments**
>
> ## Client Selection
> The client selection in online federated learning is quite an unexplored research area which is different from the main focus of this paper that is online federated model selection and fine-tuning. To clarify this in the paper, we add a discussion to the related works section to discuss client selection. This revision is highlighted in blue and can be found on page 9. Specifically, we explain that in the proposed OFMS-FT, the server selects clients for their participation in model fine-tuning uniformly at random. This choice aims to avoid differentiating among clients and fine-tune models in the favor of any clients. Nevertheless, an intriguing direction for future research is to investigate how alternative client selection strategies, beyond uniform selection, could enhance client regret in the context of online federated learning.
>
> ## Notation
> We revise the notation to address your concerns. We use the notation $\mathbb A$ for all sets. Based on your comment, we replace the notations $\beta$, $\Omega$, $\mathcal C$ and $C$ with $e$, $E$, $\mathbb C$ and $R$, respectively.
>
> ## Minor Comments
> * We revised the mentioned sentence and we omitted the “sheer volume of data” phrase.
> * In section 2, we add that “*This continues until the time horizon $T$*” to provide the formal definition of $T$ in section 2.
> * We fix the issue by revising “larger storage capacity store models” to “larger storage capacity stores models”.
> * We have revised denotes to denote in all parts of the manuscript to fix the issue.
> * We add “the” before “adversarial setting” in the paper.
> * We revised section 2 and we make it clear that the environment chooses the label through a process unknown to clients.
> * We revised the last sentence of paragraph 1 in section 3.1 to fix any grammatical issues.
> * We have included numbered equations in the paper that may not be explicitly referenced. We believe this inclusion enhances the readability and facilitates discussions about our paper, as readers can easily locate and refer to equations within the paper, even if they are not specifically cited in the subsequent text.
> * This is true that (a1) holds for $\forall k \in [K]$. We revise (a1) and we add that it holds for $\forall k \in [K]$ to make this clear.
> * We fixed the issues with references Cella et al. (2020) and Deng et al. (2020) by adding the venue.

---

> > ### Comment · Reviewer_EDLF · 2023-12-12
> > **Reply to Modifications**
> >
> > Hi authors,
> >
> > Thank you for your systematic and detailed response to my comments, and for graciously taking them on board. Having reviewed the revised manuscript, I believe it is materially improved, and my concerns have been addressed very well. I will recommend acceptance of the paper.

---

### Decision · Action_Editor_zk5N · 2024-01-12

**Recommendation:** Accept as is

**Comment:**

All three reviewers recommended an acceptance. During the discussion period, a major concern regarding the regret measure used in the paper was raised: the server’s regret can be negative since the performance of the best possible single action in hindsight could be outperformed by the performance of actions which are optimised on a per-client basis. The revised manuscript has addressed this critical issue by revising the definition of server regret. The revised manuscript has adequately addressed several questions concerning the experimental setup and client selection.

**Audience:**

All three reviewers answered positively to this question.

**Claims And Evidence:**

The paper introduces an algorithm designed for addressing the problem of online model selection (i.e. selecting an appropriate prediction model from a set of candidate models), specifically in a federated learning scenario where clients face limitations in communication and storage space. The algorithm's regret is analysed, and experimental results compare its performance to other algorithms. Baseline algorithms are not tailored for the specific task at hand, as they are designed for scenarios such as online federated learning without model selection, online model selection without federation, or federated model selection in offline settings.